# Thermodiffusive desalination

Shuqi Xu [1], Alice J. Hutchinson [1,2], Mahdiar Taheri [1], Ben Corry [2] & Juan F. Torres [1] ✉

Desalination could solve the grand challenge of water scarcity, but materials-based and conventional thermal desalination methods generally suffer from scaling, fouling and materials degradation. Here, we propose and assess thermodiffusive desalination (TDD), a method that operates entirely in the liquid phase and notably excludes evaporation, freezing, membranes, or ion-adsorbing materials. Thermodiffusion is the migration of species under a temperature gradient and can be driven by thermal energy ubiquitous in the environment. Experimentally, a 450 ppm concentration drop was achieved by thermodiffusive separation when passing a NaCl/H$_2$O solution through a single channel. This was further increased through re-circulation as a proof of concept for TDD. We also demonstrate via molecular dynamics and experiments that TDD in multi-component seawater is more amenable than in binary NaCl/H$_2$O solutions. Numerically, we show that a scalable cascaded channel structure can further amplify thermodiffusive separation, achieving a concentration drop of 25000 ppm with a recovery rate of 10%. The minimum electric power consumption in this setup can be as low as 3 Wh$_e$ m$^{-3}$, which is only 1% of the theoretical minimum energy for desalination. TDD has potential in areas with abundant thermal energy but limited electrical power resources and can contribute to alleviating global freshwater scarcity.

Worldwide water scarcity is a severe problem that has worsened in recent decades[1]. The factors exacerbating water scarcity include population growth, inefficient use of water resources, increased water usage in agriculture and industry[2] and climate change[3]. Desalination technologies could mitigate water scarcity so have been receiving significant attention since the 1960s. In 2017, the global cumulative desalination capacity exceeded 100 million m$^3$ day$^{-1}$ [1,4]. To date, all desalination methods fall into two categories: materials-based methods such as reverse osmosis (RO), electrodialysis (ED)[5] and ion-adsorbing materials[6]; and thermal-based methods such as multi-stage flash, interfacial evaporation[7] and freeze desalination[8]. Thermal methods are appealing because they are driven by low-grade thermal energy (e.g. from waste heat or solar irradiation) and hence have the potential of decentralising desalination processes. However, current thermal desalination methods suffer from scale deposition, fouling and corrosion caused by phase change[9], as well as high energy consumption at around 100 kWh$_{th}$ m$^{-3}$ [10]. Preferred for

their lower energy consumption between 3 to 7 kWh$_e$ m$^{-3}$ [11], RO has the highest installed capacity. ED recently started being developed at an industrial scale[12,13] and can be more energy-efficient than RO when the salt concentration in the feedwater is lower than ca. 5000 ppm[10], which is however much lower than the salinity of seawater (between 30,000 and 35,000 ppm). Despite being mature technologies, membrane fouling and degradation are still major issues related to all membrane methods[14]. Therefore, research interest in developing novel desalination concepts is growing[4], e.g. capacitive deionisation[15], ion-concentration polarisation[16] and contactless steam generation[17]. Methods beyond the scope of desalination, e.g. atmospheric water harvesting[18], are also receiving growing attention. Although most emerging technologies have not reached the maturity required for commercialisation because of their low throughput and high materials cost, their advantages such as decentralised desalination and use of low-grade thermal energy have justified further development.

[1]ANU HEAT Lab, School of Engineering, The Australian National University, Canberra, ACT, Australia. [2]Research School of Biology, The Australian National University, Canberra, ACT, Australia. ✉e-mail: felipe.torres@anu.edu.au

Here, we re-visit a 150-year-old problem[19], i.e. can thermodiffusion be used as an effective means for desalination? Thermodiffusion refers to species migration due to temperature gradients and is also termed the Soret effect or thermophoresis for large particles. Thermodiffusion is ubiquitous in nature and is thought to be related to many important phenomena including possibly the origin of life[20,21]. However, it has never been used as means for desalination, despite being a phenomenon first described in detail by Soret in 1879 through the observation of concentration inhomogeneities in a non-isothermal aqueous solution of NaCl[19]. There are only two theoretical papers briefly suggesting the possibility of thermodiffusion-based desalination[22,23]. Nonetheless, thermodiffusion has found use in few, yet essential, engineering applications, such as uranium enrichment in the Manhattan Project[24], prediction of hydrocarbon distribution in oil reservoirs[25,26] and analysis of biomolecular interactions[27]. In addition, many applications have been proposed based on thermodiffusion including nanoscale light tweezers[28], carbon capture[29], hydrogen separation[30] and microfluidics-based separation of binary mixtures[31–34]. However, high-throughput applications based on thermodiffusion, such as thermodiffusive desalination (TDD), are yet to be achieved.

Thermodiffusion can be understood by considering the mass flux equation. In the presence of both concentration and temperature gradients, the mass flux in a binary solution is given by

$$\mathbf{J} = -\rho D \nabla C - \rho C (1 - C) D_T \nabla T, \qquad (1)$$

where the first term on the right-hand side of the equation describes the isothermal (or Fickian) mass diffusion due to a concentration gradient $\nabla C$, and the second term describes thermodiffusion or the transport of species due to a temperature gradient $\nabla T$. In Eq. (1), $D$ is the Fickian diffusion coefficient which is always positive because the molecules diffuse spontaneously from high to low concentration in a quasi-isothermal condition due to Brownian motion. In contrast, $D_T$ is the thermodiffusion coefficient which can be either positive (thermophobic) or negative (thermophilic), depending on thermal preference of the species[35]. Seawater is a multi-ion aqueous solution, but a quasi-binary approximation could be relatively accurate due to electroneutrality[36]. The Soret coefficient $S_T$ is defined as the ratio between the thermodiffusion and Fickian diffusion coefficients, $S_T \equiv D_T/D$. Based on Eq. (1), $\Delta C$ can be approximated as $\Delta C \approx -C_0(1 - C_0)S_T\Delta T$ for quasi-linear temperature and concentration profiles with small changes. $S_T$ of NaCl in water is in the order of $10^{-3}$ K$^{-1}$ [37,38], meaning the $\Delta C$ that can be induced by thermodiffusion is small, limited by the $\Delta T$ achievable within the liquid phase of seawater. Based on measured values of $S_T$, a temperature difference of 40 K yields $\Delta C_{walls} = 2400$ ppm. between the top and bottom boundaries for an initial concentration of $C_0 = 30,000$ ppm, i.e. 8% of $C_0$. Therefore, thermodiffusion of NaCl in water can barely overcome Fickian diffusion, and this is likely why thermodiffusion has not been implemented as the separation principle in high-throughput applications. However, TDD is promising in that it is a moderate-temperature process that can be driven by temperatures much less than 100 °C extractable from the surrounding environment, e.g. from waste industrial heat or solar thermal energy. The heat being dissipated by cooling towers or sea currents could be used for desalination. Moreover, TDD is a completely materials-free separation process that relies on a simple concept. Note that thermodiffusion is stronger at higher concentrations with a peak efficiency when the mass fraction of the species is 0.5, as per thermodiffusion mass flux in Eq. (1). This suggests thermodiffusion could be a promising brine treatment method or precursor to existing materials-based desalination methods such as RO, ED, and emerging technologies such as ion-adsorbing materials[6], which are all known to be more efficient with a lower-salinity feedwater.

In this work, we explore thermodiffusion as means of an effective water desalination method and reflect on its challenges and prospects. The method is termed thermodiffusive desalination, or TDD. First, we provide experimental results showing that thermodiffusion is able to achieve a tangible desalination level with a salinity reduction of 450 ppm and 700 ppm from an initial NaCl concentration of 30,000 ppm (seawater) and 60,000 ppm (brine), respectively. Importantly, the experimental setup is a simple parallel-plate channel with laminar flow, yet the concentration drop is relatively small after a single pass. As a proof of concept for TDD, re-circulation (multi-pass flow) through the same channel is implemented to increase the concentration drop to more than 2000 ppm. The multi-pass experiment was also performed using a substitute of seawater (i.e. a multi-ion aqueous solution) in which thermodiffusive separation of the cations was experimentally observed and then corroborated via molecular dynamics simulations. Finally, we propose and theoretically assess a cascaded structure, i.e. single flow pass through a multi-channel device, as a solution for scaling up TDD. We show a theoretical concentration drop exceeding 25,000 ppm with a recovery rate of 10%, which is amenable for high-throughput desalination. While still a proof of concept, this study provides valid pathways towards realising a single-phase thermal desalination process, demonstrating that water desalination is possible without the need for materials or phase change. Current materials-based methods utilise functional materials, such as ion-adsorbents[6], selective membranes[39] and permeable solar absorbers[7], but these methods have intrinsic issues with fouling and regeneration. Avoiding the phase changes inherent in current thermal desalination methods reduces the relatively large latent heat requirements, giving TDD the potential to be less energy intensive without the cumbersome maintenance of functional materials.

## Results
### Thermodiffusive desalination concept

The proposed concept of TDD follows the configuration depicted in Fig. 1a. A multi-component salt solution passes through a rectangular channel with a vertical positive temperature gradient (heated from the top), as shown in Fig. 1b. This channel could be a simple, single channel −as in the thermodiffusive desalination unit (TDU) depicted in Fig. 1c− or a more complex multi-channel device (section "Burgers cascade"). Note that the species in this solution can be either thermophobic or thermophilic, depending on the local temperature and concentration. Salt ions in aqueous solutions are generally thermophobic above a temperature threshold called the inversion temperature $T_0$ (ca. 12 °C for NaCl)[37]. In the designed TDU (Fig. 1c), a fully-developed laminar flow between thermostatically-controlled parallel plates is ensured, as transient flows above the critical Reynolds number induce mixing. A temperature difference is applied across the channel height $h$, with the top being heated and the bottom cooled to obtain a thermally stratified condition (otherwise, Rayleigh–Bénard instability could stir the flow[40]). In thermophobic transport, ionic species tend to accumulate towards the cold bottom side, resulting in a negative concentration gradient. Together with the negative density gradient brought by thermal stratification, species stratification ensures that natural convection-induced mixing does not occur in our thermodiffusive separation channel[35]. Heating can be achieved with low-grade heat including solar thermal energy harvested with solar absorbers[41] or industrial waste heat. Cooling can be achieved through convection with the low-temperature saline water reservoir. The thermodiffusive separated solution is bifurcated into two streams at the end of the rectangular channel using a sharp spacer. In Fig. 1c, a more detailed view of the TDU with spacer (in the inset) is shown (see photos in Supplementary Fig. 1). Here, the volumetric flow rate $Q$ of the saline feedwater is controlled with a peristaltic pump. The water is degassed before entering the TDU to avoid microbubble-induced mixing. Two

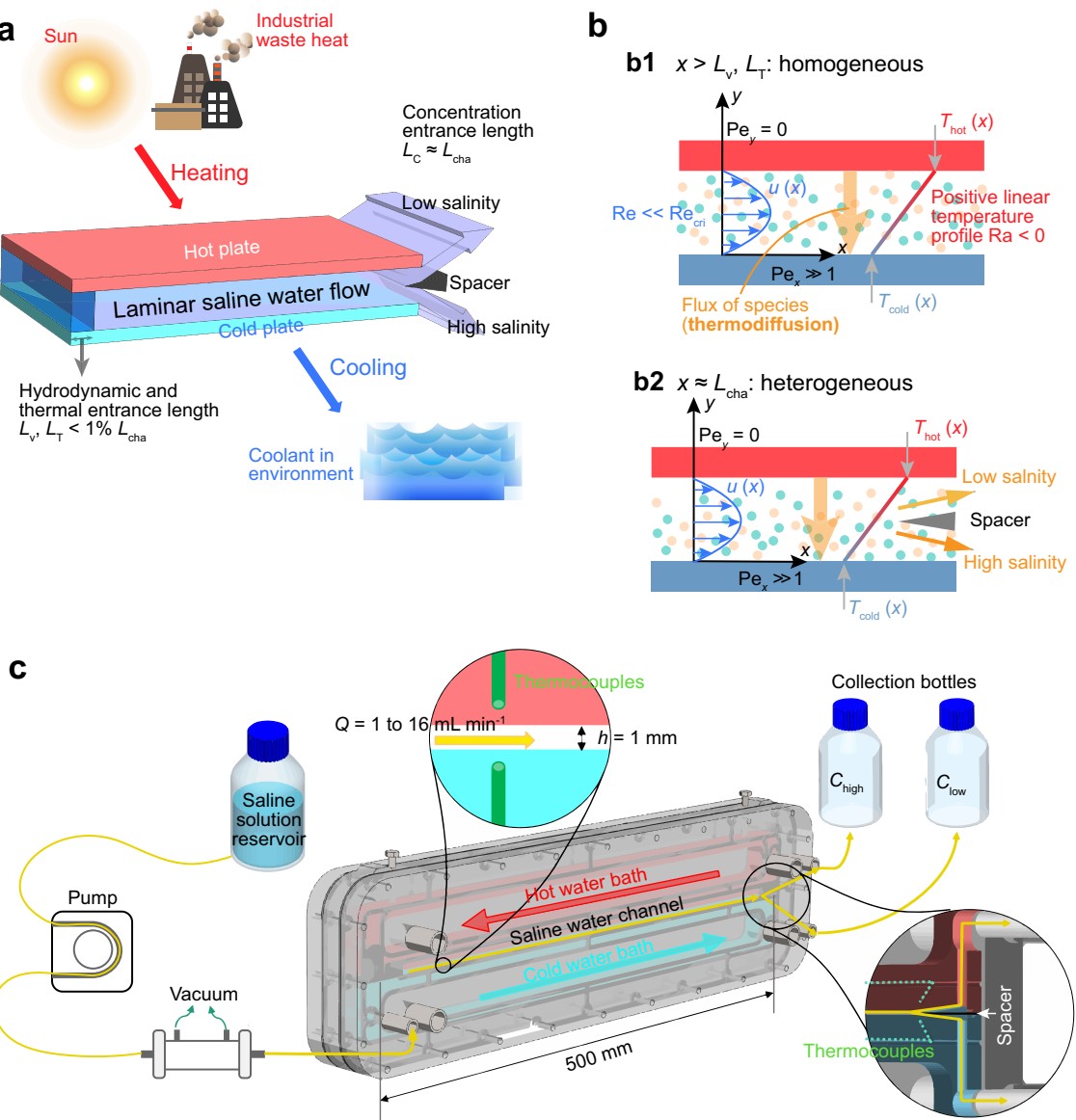

**Fig. 1 | Concept of thermodiffusive desalination and unit design. a** Concept figure showing a laminar flow of saline water passing through a thermodiffusive separation channel. The temperature difference $\Delta T$ can be established with low-grade thermal energy. For temperatures above the inversion temperature, the thermophobic ions in the solution migrate towards the cold side as the saline flow progresses along the channel. This results in an upper stream having a lower salinity than the feedwater. The hydrodynamic and thermal entrance lengths, $L_v$ and $L_T$, are both less than 1% of the total channel length, $L_{cha}$. **b** The Reynolds number, Ra, indicates that $L_v$ and $L_T$ are negligible compared to $L_{cha}$. Thus, the flow in the thermodiffusive separation channel can be approximated as a laminar, fully-developed planar Poiseuille flow with a positive quasi-linear temperature profile. $x$ is the saline water flow direction and $x = 0$ corresponds to the channel inlet. The Péclet number, Pe, indicates an advection-dominant mass transport in the $x$ direction and diffusion-dominant transport in the $y$ direction. At the inlet, the solution is homogeneous, whereas at the outlet it becomes heterogeneous due to thermodiffusion. **c** Thermodiffusive desalination unit (TDU) design. The volumetric flow rate $Q$ of the saline mixture in the TDU is controlled by a peristaltic pump between 1 to 16 mL min⁻¹. Feedwater is degassed before entering the channel. The fluid path for the saline water is indicated by yellow lines. The channel is 500 mm long, 20 mm wide, and 1 mm high. At the exit of the channel, the saline water is separated into two streams by a spacer, as shown in the inset. After bifurcation, the two streams are collected in bottles at the same height, ensuring equal pressure and hence equal flow rate. Two hollow copper blocks with water circulation create the $\Delta T$. Thermocouples were embedded in top and bottom channel walls to monitor the temperature (Supplementary Fig. 7).

hollow nickel-plated copper blocks are thermostatically controlled with a counter-flow water circulation system to establish a positive temperature gradient throughout the channel. Two bottles are placed at the same height so that equal flow into each bottle is ensured.

### Single-pass TDD
Since thermodiffusion in multi-component solutions is not adequately understood and there are limitations in the accurate measurement of diffusion in multi-component mixtures[42,43], we started our investigation with a single-pass TDD experiment using the most common binary

approximation of seawater: a NaCl/H₂O solution. We recently reported numerical simulation results[22] of the thermodiffusive separation within a parallel-plate channel with a plane Poiseuille flow and a linear temperature gradient. Details of the numerical simulations relevant to this work are available in the Supplementary Method 1, along with Supplementary Figs. 2–5. The design rationale for the TDU is detailed in Supplementary Method 2, along with Supplementary Fig. 6. The flow within the TDU is laminar (Re < 350) because of the reduced flow speed required to achieve a nearly fully developed concentration profile at the channel outlet, i.e. a rather long resilience time ($t > 1$ min) within

our 0.5 m channel is needed to obtain a nearly complete thermo-diffusive separation. With our TDU design ($h$ = 1mm) and target flow rates (1 – 16 mL min$^{-1}$), the hydrodynamic, thermal, and concentration entrance lengths are around 0.1%, 1%, and 100% of the entire channel length, respectively. We also found that the mass transport is advection-dominant along the channel (Péclet number $Pe_x \approx 10^3$) while diffusion and thermodiffusion dominate across the channel height ($Pe_y = 0$).

The experimental and numerical results are shown in Fig. 2a, where the concentration difference $\Delta C$ (between top and bottom solutions in the collection bottles of Fig. 1c) is shown as a function of the measured temperature difference $\Delta T_{meas}$ along the channel (measured with thermocouples). A typical temperature profile along the channel is shown in Supplementary Fig. 7b. The mean temperature $T_{mean}$ and temperature difference between walls $\Delta T$ were chosen such that the temperature of the bottom wall $T_{cold}$ in the TDU (either single

channel or a Burgers cascade) is above the inversion temperature at which thermophobic–thermophilic transition occurs. The inversion temperature has been reported to be between 10 °C[38] and 12 °C[37] for an aqueous NaCl/H$_2$O at seawater concentration. Measurements and modelled predictions of $\Delta C$ were conducted via phase-shifting inter-ferometry (PSI)[44] and computational fluid dynamics (CFD)[22], respectively. An accurate measurement of the $\Delta C$ is critical in assessing the performance of TDD, and we found that our in-house PSI was more accurate than commercial salinity meters based on electrical resis-tance. Supplementary Method 3 and Supplementary Fig. 8 provide details on the PSI measurement and associated uncertainty. The ver-tical error bars in Figs. 2a, b and 3a, b depict the errors $\delta\Delta C$ in con-centration difference $\Delta C$ measurements, and the errors $\delta C_{drop}$ in concentration drop $C_{drop}$ measurements, respectively. Measurement errors mainly arise when extracting the phase difference from unwrapped phase-shifted data (insets of Fig. 2a) between $C_{high}$ and $C_{low}$

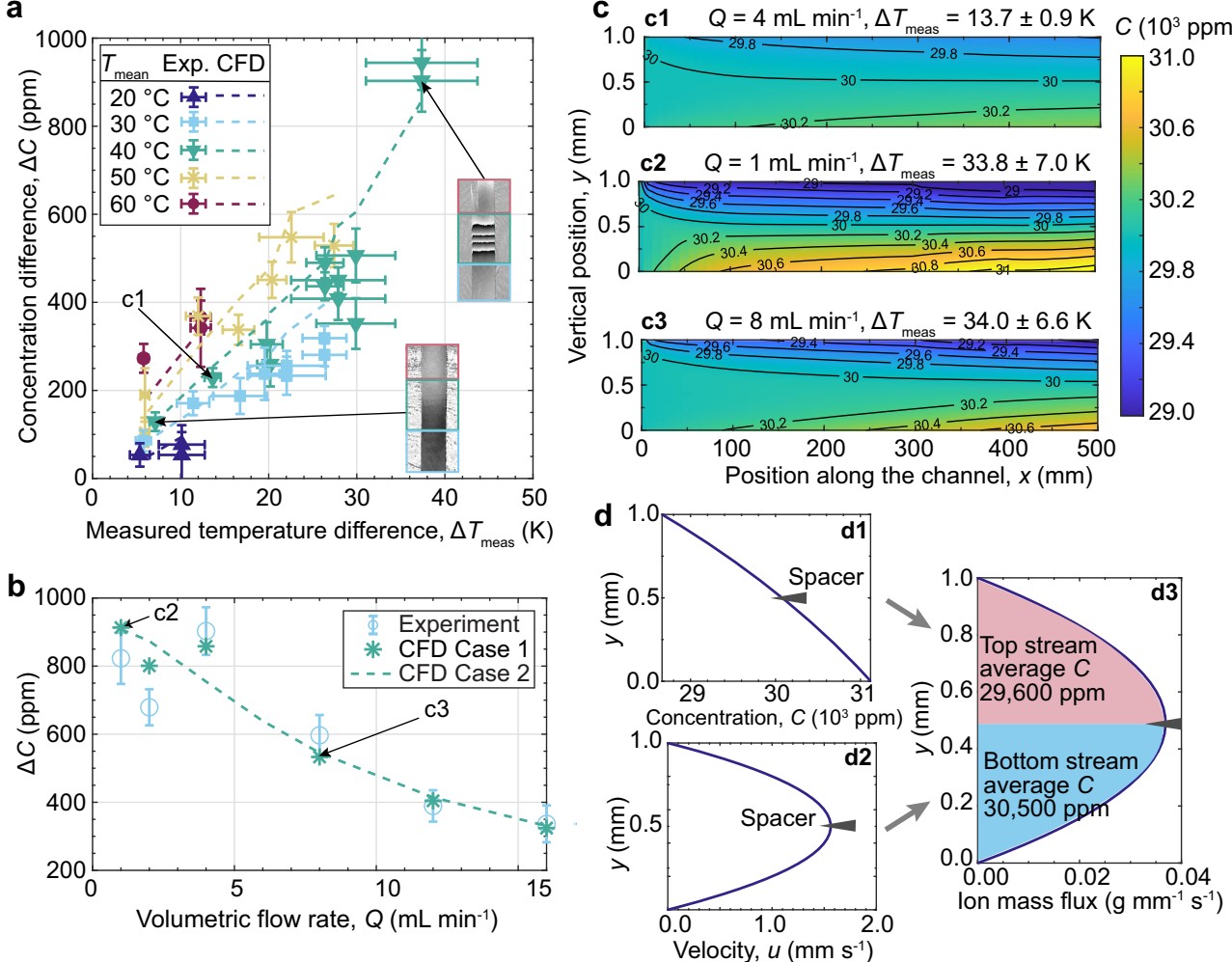

**Fig. 2 | Performance assessment of single-pass thermodiffusive desalination. a** The concentration difference $\Delta C$ between the top and bottom collection bottles as a function of the average measured temperature difference along the channel $\Delta T_{meas}$ for different mean temperatures $T_{mean}$. The experimental results (solid markers) are plotted with horizontal and vertical error bars representing the standard deviation of $\Delta T$ measurements (with six thermocouple pairs) and the uncertainty in $\Delta C$ measurements (with phase-shifting interferometry, PSI). The insets are PSI images following the experimental procedure reported in[35,44]. CFD results (dashed lines) with experimental temperature values as boundary condi-tions are included. Monotonic dependencies are revealed. **b** The effect of flow rate $Q$ on $\Delta C$ is reported for $T_{mean} \approx 41$ °C and $\Delta T_{meas} \approx 34$ K both experimentally and numerically. The errors in the experimental $\Delta C$ are standard deviations from PSI

measurements. CFD Case 1 uses the thermocouple temperatures measured in each experiment as boundary conditions, whereas CFD Case 2 uses the common wall temperature profile of the experiment with lowest volumetric flow rate of 1 mL min$^{-1}$. $\Delta C$ remains roughly the same when $Q < 5$ mL min$^{-1}$. The effect of slight variation in the temperature field on the separation is prominent for small $Q$. **c** CFD Case 1 concentration contours for different $\Delta T_{meas}$ and $Q$. Larger $\Delta C$ are observed for greater $\Delta T_{meas}$ and smaller $Q$. **d** CFD computation of the concentration dif-ference between collection bottles. Temperature-dependent $S_T$ yields a quasi-linear concentration profile within the cell (**d1**), together with the parabolic velo-city profile (**d2**), yields the ion mass flux as a function of the height (**d3**). Dividing the integrated ion mass flux in the top and bottom halves with the corresponding flow rate yields the concentration at the collection bottles.

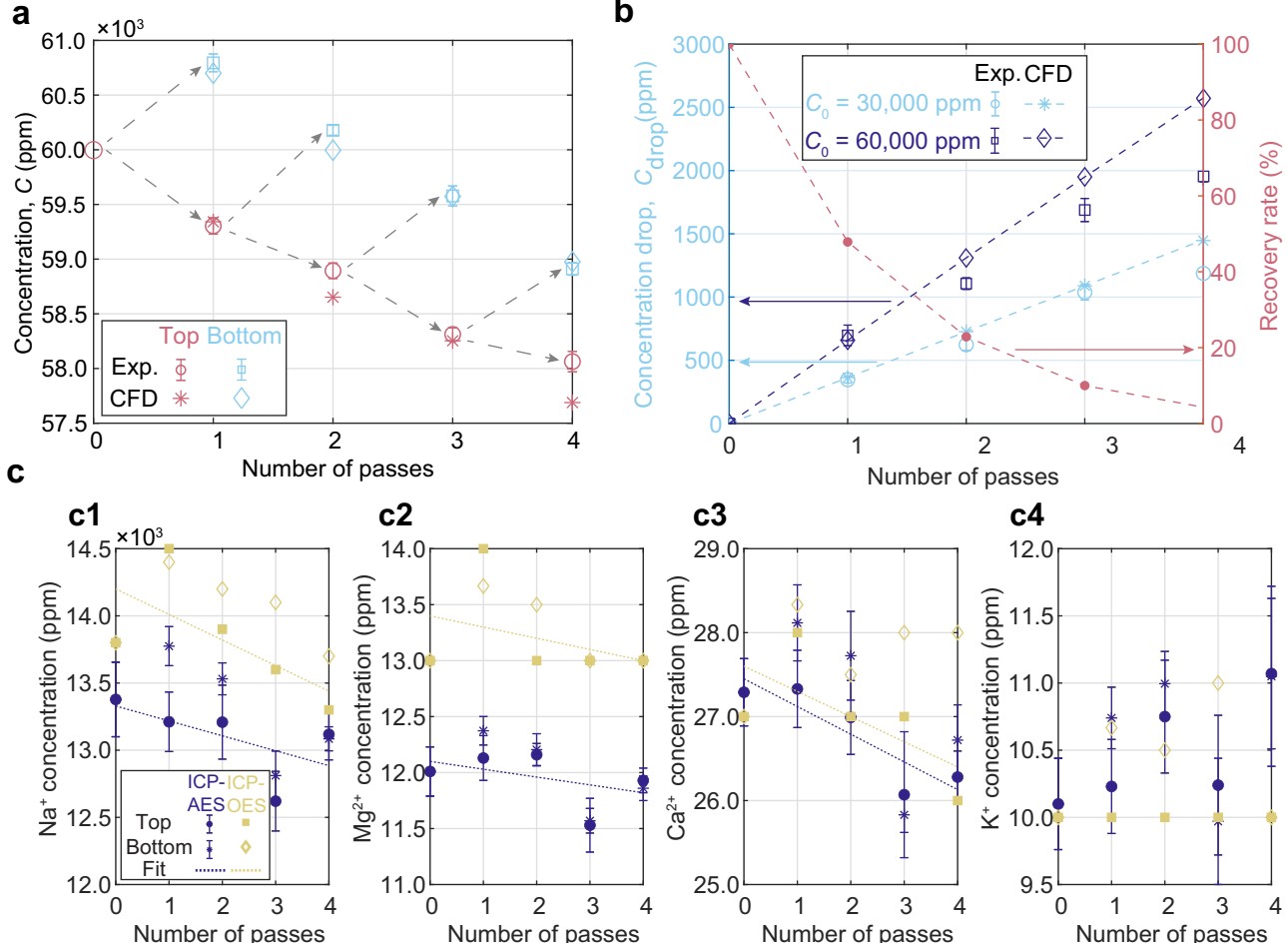

**Fig. 3 | Multi-pass thermodiffusive desalination for aqueous NaCl and seawater.** Both seawater level and brine level (double the seawater concentration) are considered. **a** The initial concentration of the NaCl in aqueous solution was 60,000 ppm. NaCl concentration in both the top and bottom streams are plotted for each pass. Both experimental results and numerical results are demonstrated. The errors in the experimental $\Delta C$ are standard deviations from PSI measurements. In each pass, the top stream with a lower salinity was accumulated and put into the next pass. The NaCl concentration in the top stream kept dropping, demonstrating that thermodiffusion is scalable beyond a single pass. **b** The concentration drop is plotted as a function of number of passes recirculated for two different initial concentrations (30,000 and 60,000 ppm). On the left axis, the concentration drop $C_{drop}$ achieved after four passes were 1200 ppm and 2000 ppm for an initial

concentration of 30,000 and 60,000 ppm, respectively. On the right axis, the recovery rate is shown. There is a trade-off between $C_{drop}$ and recovery rate. **c** Same experiment was performed using multi-ion seawater substitute. The concentration of different cations: $Na^+$ (**c1**), $Mg^{2+}$ (**c2**), $Ca^{2+}$ (**c3**), and $K^+$ (**c4**), was measured by two different ICP methods for the same set of samples collected in each pass. Linear fits were performed for the top stream concentrations to show the concentration reduction trend. The errors in ICP-AES measurements are the standard deviations between five replicates. The errors of the ICP-OES results were not available. $Na^+$ is most abundant in quantity and provides most reliable results. For ICP-AES results, the $C_{top} = (−190n + 14{,}200)$ ppm, i.e. $C_{drop} = 190$ ppm and $C_0 = 14{,}200$ ppm. For ICP-MS results, $C_{top} = (−111n + 13{,}329)$ ppm. Linear fits are also applied to $Mg^{2+}$ and $Ca^{2+}$. For $K^+$, however, no trendline is fitted due to lack of discernible trend.

solutions (i.e. those extracted from the collection bottles in Fig. 1c). A typical relative PSI-based error $\frac{\delta \Delta C}{\Delta C}$ is around 10% for the relatively small salinity difference $\Delta C < 1000$ ppm. In contrast, a commercial electrical resistance-based salinity meter may have a relative error exceeding 100% since the measured $\Delta C$ is very small. The CFD model takes the measured temperature along the top and bottom walls of the channel (Supplementary Fig. 7) as boundary conditions and considers temperature-dependent Soret and Fickian diffusion coefficients, while assuming a fully developed flow between the plates.

For the same mean temperature $T_{mean}$, we could confirm that $\Delta T_{meas}$ is roughly proportional to $\Delta C$, which is an expected result according to Eq. (1). Insets of Fig. 2a show clear fringe patterns between the two outlet samples, which is evidence of thermodiffusive separation of NaCl in the channel. Under a larger $\Delta T_{meas}$, more fringes in the phase-shifted data are indicative of a larger $\Delta C$ between the top and bottom solutions. Furthermore, the good agreement between PSI and CFD results confirm that no mixing occurred in our channel (as the latter assumes a fully-developed laminar flow). Note that, for the same

$\Delta T_{meas}$, $\Delta C$ increases with mean temperature $T_{mean}$. This is expected as the Soret coefficient of aqueous NaCl is known to increase monotonically with temperature[37].

The effect of volumetric flow rate $Q$ was also analysed. Figure 2b shows that a near-maximum $\Delta C$ was achieved for $Q < 5$ mL min⁻¹. This is because the saline water from the inlet remains in the 1 mm high channel for more than 2 min, which is enough time for full separation to occur. Larger flow rates result in a drop of $\Delta C$ as the residence time of the solution in the channel is not long enough for thermodiffusion to fully separate NaCl before reaching the spacer. Furthermore, different water bath flow configurations yield different wall temperature profiles (Supplementary Fig. 7). A counter flow configuration with the hot water bath flowing in the opposite direction to the saline water was chosen (Supplementary Method 4). Bubbles from the water circulation may accumulates at the heat exchange surface (Supplementary Fig. 1b), which reduce the local heat transfer coefficient and hence lower $\Delta T$. Without visible bubbles, the $\Delta C$ obtained for the same set water circulation temperatures is within the PSI measurement

accuracy, meaning an excellent repeatability was obtained. However, despite aiming at having the same wall temperature profiles when investigating the effect of the volumetric flow rate $Q$, the wall temperature varied due to the different thermal resistance caused by the bubbles. Difference in wall temperature slightly change predictions of $\Delta C$ for CFD Case 1 and Case 2, as shown in Fig. 2b when the volumetric flow rate is less than 5 mL min⁻¹. CFD Case 1 was performed using measured wall temperature profiles (for the corresponding $Q$) as boundary condition, while CFD Case 2 was performed using the wall temperature profile obtained in the $Q = 1$ mL min⁻¹ experiment. The impact of slight variations in temperature profiles on the $\Delta C$ is more obvious for small $Q$. In Fig. 2c (CFD Case 1), concentration profiles on the vertical plane along the channel are shown for three sets of $Q$ and $\Delta T_{meas}$. As expected, a smaller separation is obtained for a smaller $\Delta T_{meas}$ (Fig. 2c1 vs. Fig. 2c2, c3). Furthermore, when a very small $Q = 1$ mL min⁻¹ is implemented (Fig. 2c2), the concentration profile becomes fully developed halfway through the channel (slight variations are still seen in Fig. 2c2, but these are due to non-uniform wall temperatures as shown in Supplementary Fig. 7). In contrast, a larger $Q$ increases the concentration developing length (Fig. 2c3).

The calculation procedure for obtaining the concentration at the top and bottom collection bottles from CFD results is depicted in Fig. 2d. We show that despite the seemingly large $\Delta C_{walls}$ between the top and bottom boundaries, due to the parabolic velocity profile, the $\Delta C$ between the top and bottom collection bottles is significantly smaller than in a convectionless environment. The width of the TDU channel is $w$ and the top stream concentration is calculated as $\frac{\rho w}{Q/2} \int_{h/2}^{h} C(y)u(y)\mathrm{d}y$, whereas the bottom stream concentration as $\frac{\rho w}{Q/2} \int_{0}^{h/2} C(y)u(y)\mathrm{d}y$. The $\Delta C$ between top and bottom streams is around 37% of that between the boundaries. Hence, for $\Delta C = 900$ ppm as in Fig. 2a, the $\Delta C_{walls}$ between top and bottom boundaries is around 2400 ppm.

## Multi-pass TDD

The maximum $\Delta C$ after a single pass (Fig. 2a) is ca. 900 ppm for $\Delta T_{meas} = 37$ K. However, 900 ppm is only 3% (relative value) of the initial NaCl concentration in seawater, ca. 30,000 ppm. Moreover, the salinity reduction or concentration drop $C_{drop}$ of the top stream concentration from the inlet value is only half of $\Delta C$ between top and bottom streams (i.e. 450 ppm). This small drop is not enough to obtain low-salinity water useful for agriculture, which may require 95% relative salinity reduction and accounts for 69% of water use worldwide[45]. To amplify thermodiffusive separation, a simple way is to accumulate the top low-concentration stream in a container and then pass it again through the same channel. This re-circulation experiment was done for two initial NaCl feedwater concentrations, seawater level (30,000 ppm) and brine level (60,000 ppm). The concentration of NaCl in the top and the bottom streams after each pass is shown in Fig. 3a for brine feedwater. Note that the bottom high-concentration stream is discarded after each run, while the top low-concentration stream is re-circulated. CFD Case 1 results have a very good agreement with the experiments, except for the last pass as the recovered volume was low and contamination could have affected the experimental result.

Figure 3b shows the concentration drop $C_{drop}$ increasing with the number of passes, demonstrating that TDD can be scaled up beyond the modest results obtained from a single pass. Furthermore, it is noticed that $C_{drop}$ is larger in brine than in seawater, which can be attributed to the larger thermodiffusive mass flux at increased concentration following Eq. (1). This feature suggests that TDD may be a potential pre-treatment method for high-salinity feedwater. However, the significant drawback of this multi-pass TDD is that the volume of the desalinated stream drops exponentially, halving after each pass. In the fifth pass, the recovery rate was ca. 4%, which is excessively small for a yield with relatively large salinity, ca. 95% of its initial value.

Therefore, another method for amplifying thermodiffusive separation should be devised for it to be amenable in desalination applications.

Another factor that may affect TDD in real applications, is that seawater is a multi-component ionic solution and some ions may hinder thermodiffusive separation if their thermodiffusive transport opposes or is weaker than that of the thermophobic ions of Na⁺ and Cl⁻ (the major species in seawater). Despite being discovered more than 150 years ago, the mechanisms behind thermodiffusion are still unclear and quantitative measurements have been limited to ternary mixtures[46]. Here, we are more interested in the 'collective' thermodiffusive transport of all the seawater components assuming that electrostatic interactions have them diffuse as a single effective species. To assess TDD in seawater, natural sea salt was dissolved in deionised water at a concentration of 30 g L⁻¹ to prepare an artificial substitute of seawater. Figure 3c shows the multi-pass experimental result with seawater substitute. Inductively coupled plasma atomic emission spectroscopy (ICP-AES) and inductively coupled plasma mass spectrometry (ICP-MS) were used to measure the concentration of different cations in the same set of samples collected. Although the exact composition of the seawater substitute was not the same as in natural seawater reported in the literature[10], it demonstrates that the TDD principle applies to multi-ion aqueous solutions.

A linear fit was applied to the top solution concentrations to determine the concentration drop $C_{drop}$ in each pass. Based on the numerical analysis in Fig. 2d, $C_{drop}$ can be translated back to the concentration profile across the channel height to calculate the Soret coefficient $S_T$. The process to derive $S_T$ from $C_{drop}$ is detailed in the Supplementary Method 5. $S_T$ was $2^{+1}_{-1} \times 10^{-3}$ K⁻¹ based on both the ICP-AES and ICP-MS measurement results. The superscripts and the subscripts originates from the process of performing a linear fit to $C_{drop}(n)$ (Fig. 3c). For the linear fit, the slope is the concentration drop per pass $C_{drop}$ and the intersection at $n = 0$ is the initial concentration $C_0$. When assuming a planar Poiseuille flow (Fig. 2d2), both $\Delta C_{walls}$ and the concentration profile $C(y)$ (Fig. 2d1) can be calculated from $C_{drop}$ (Fig. 2d3). With the known $C(y)$, $S_T$ can be calculated as per Supplementary Method 5. The superscripts and subscripts correspond to the linear fit with the largest and smallest slopes with a confidence interval of 95%, respectively. These values are comparable to or greater than that of the binary NaCl/H₂O. Although an accurate measurement of $S_T$ from ICP analyses is challenging, we can conclude that TDD is not hindered by the multi-component nature of seawater.

## TDD at the molecular level

Molecular dynamics (MD) simulations were performed on NaCl and multi-ion seawater substitute solutions to assess the application of the TDD principle to multi-component electrolytes. The TDU experimentally achieved a salinity reduction in both NaCl (Fig. 3a, b) and seawater substitute solutions (Fig. 3c). However, there was a large uncertainty in the reported salinity measurements using ICP methods. Therefore, simulations at the molecular level are sought to confirm the experimental observation that thermodiffusive separation occurs in a multi-component ionic solution of seawater. Furthermore, MD simulations may also provide insight into ion–ion and ion–water interactions, as well as providing a single theoretical framework to compare thermodiffusion in artificial seawater substitute and the simpler binary NaCl/H₂O approximation.

The MD simulation results shown in Fig. 4 predict seawater ions exhibit a thermophobic behaviour (see Supplementary Method 6 and Supplementary Figs. 9–11 for more details on methods and MD results). The simulation was run with an average temperature of 40 °C and a temperature difference of 40 K to match the optimal operating conditions of the TDU and maximise the expected concentration drop (see Fig. 2a). The total concentration of the multi-ion seawater model and NaCl solutions was more than doubled (i.e. brine) to improve the counting statistics of the MD results. The exact

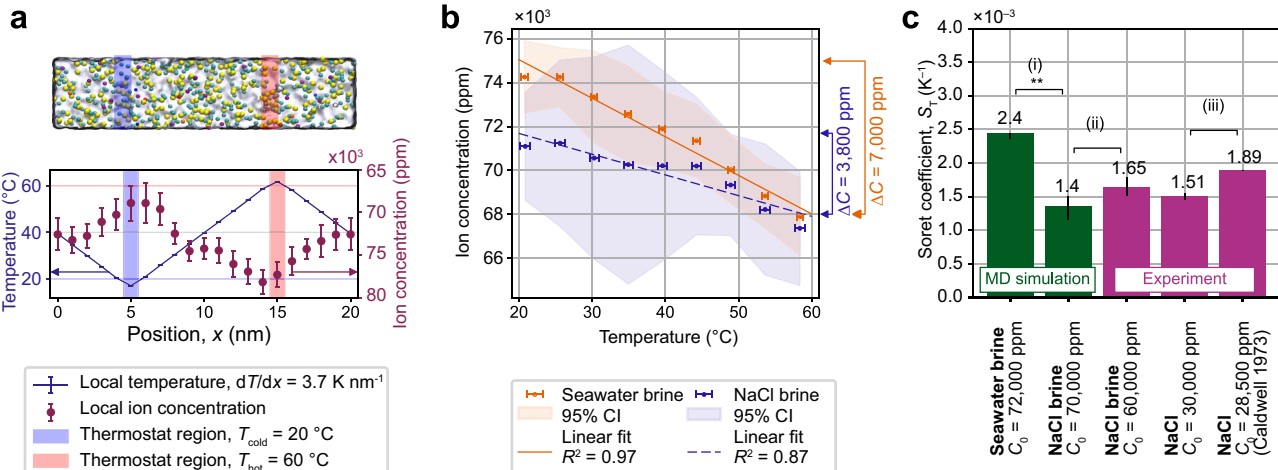

**Fig. 4 | Molecular dynamics modelling of thermodiffusive desalination.**
**a** Molecular dynamics (MD) model of seawater brine under a non-uniform temperature profile spanning ca. 20–60 °C. The ion concentration (for all ions) decreases as the temperature increases, demonstrating thermophobic behaviour of seawater ions in silico. The simulation volume (top) contains water molecules (light grey), and seawater ions (multiple colours). A cold and hot thermostat of width 1 nm are placed at $x = 5$ nm and $x = 15$ nm (highlighted regions). The steady state temperature (solid line) and time-averaged ion concentration (markers) between $[t_0 = 20, t_{\text{final}} = 280]$ ns of simulation time are plotted. The error bars are the standard deviations for the time-averaged temperature and concentration within the steady state simulation time. **b** Average ion concentration in simulated seawater brine and NaCl brine across temperature. The concentration drop for seawater brine ($7000 \pm 400$ ppm) was larger than concentration drop for NaCl brine ($3800 \pm 600$ ppm). The ion concentration for seawater brine (in orange) and NaCl brine (in blue) is averaged over four and three independent replicates, respectively, and shown with standard error in the mean of the local temperature (markers with error bars). The spread in the average ion concentration is shown at a 95% confidence interval (shaded regions) due to the small number of replicates. A linear line of best fit is reported with an $R^2$ value for seawater brine (solid line) as $C(T) = (-176 \pm 11)T + (78{,}600 \pm 500)$ ppm and NaCl brine (dashed line) as $C(T) = (-95 \pm 14)T + (73{,}600 \pm 600)$ ppm. **c** Comparison of Soret coefficients obtained from MD simulations and experiments (both from this work and literature). (i) The Soret coefficient calculated for seawater brine is 1.8 times larger than that of NaCl brine. The errors arise from the $S_T$ calculation process. When applying different possible fits to the results (with a confidence interval of 95%), they result in different $\Delta C$. Details of the error derivation are available in Supplementary Method 5. The double asterisk annotation (**) indicates that the increase in the Soret coefficient is statistically significant with $p = 0.003$ (i.e. $p < 0.01$), reported by a one-sided $t$-test. (ii) The Soret coefficient for NaCl brine obtained from MD simulations is comparable to that from experiments. (iii) The experimental Soret coefficient for NaCl in this work is lower than that of ref. 37.

compositions of the seawater brine is available (Supplementary Table 1). The tested model solutions are therefore referred to as seawater brine and NaCl brine. The average concentration of seawater ions shown in Fig. 4a increases as the temperature in the simulation decreases, demonstrating that the thermodiffusion of seawater ions is thermophobic in the temperature range 20–60 °C.

MD simulations predict that the modelled multi-component seawater had a greater salinity reduction than that in the binary NaCl/H₂O system. Both brine of seawater and NaCl behaved thermophobically, with ion concentration decreasing with increasing temperature, as shown in Fig. 4b. Seawater brine had a $\Delta C_{\text{walls}}$ of $7000 \pm 400$ ppm between the hot and cold boundaries, larger than the $3800 \pm 600$ ppm $\Delta C_{\text{walls}}$ predicted for NaCl brine. A comparison of the Soret coefficients in Fig. 4c(i) shows that the predicted Soret coefficient of $2.4^{+0.0}_{-0.1} \times 10^{-3} \text{K}^{-1}$ for seawater brine was $1.8^{+0.3}_{-0.2}$ times larger than the predicted Soret coefficient of $1.4^{+0.2}_{-0.2} \times 10^{-3} \text{K}^{-1}$ for NaCl brine. To relate the scatter plot as in Fig. 4b with $S_T$, a linear fit was performed to obtain a continuous concentration profile $C(y)$. $S_T$ is then calculated by finding the value of $S_T$ that renders an analytical solution of $C(y)$ close to the linear fit. The linear fit with the largest and smallest slopes corresponds to the superscript and subscript in the reported Soret coefficient, respectively. Details of the analytical process can be found in Supplementary Method 6. Analysis of the individual ion species concentration profiles suggest that the larger separation predicted for seawater brine is caused by the stronger thermodiffusive separation of Mg²⁺ and the consequent electrostatic interaction between Mg²⁺ and the other ions (Supplementary Fig. 11). Overall, the MD simulations predicted that thermodiffusion can produce a greater salinity reduction in our multi-component seawater brine model than in NaCl brine.

Confidence in the enhanced thermodiffusive response of multi-component seawater brine is gained from the agreement between MD simulation predictions and experimental measurements of NaCl brine. The $3800 \pm 600$ ppm $\Delta C_{\text{walls}}$ of NaCl brine predicted by simulation (reported in Fig. 4b) is larger than the $3100 \pm 160$ ppm $\Delta C_{\text{walls}}$ of NaCl brine measured experimentally in the TDU (reported in Supplementary Table 2) for comparable concentrations. However, note that the MD simulation has a temperature difference of 40 K, and the comparable NaCl experiment achieved a reduced temperature difference of ca. 31 K (Supplementary Table 2). At this reduced temperature difference, the MD result predicts a $\Delta C_{\text{walls}}$ of $2900 \pm 400$ ppm (calculated from the best linear fit for NaCl brine in Fig. 4b), better agreeing with the TDU-measured $\Delta C_{\text{walls}}$. The Soret coefficient of NaCl brine predicted by simulation is $1.4^{+0.2}_{-0.1} \times 10^{-3} \text{K}^{-1}$, a slight underestimate compared to the experimentally determined Soret coefficient of $1.65^{+0.08}_{-0.14} \times 10^{-3} \text{K}^{-1}$, as reported in Fig. 4c(ii). However, the estimates are within their error ranges. Experimental values for the Soret coefficient found in this work are lower than that of the literature value for the Soret coefficient published in[37], reported in Fig. 4c(iii). Overall, the simulation predictions for NaCl brine agree with experimental measures of salinity reduction and Soret coefficient made in this work, and are less compared to early literature values[37]. As there are no comparable values for the Soret coefficient of the multi-ion seawater substitute published in the literature, validation of the NaCl brine Soret coefficient gives confidence to the enhanced thermodiffusive response predicted for seawater brine.

### Burgers cascade
The simple multi-pass re-circulation scheme (Fig. 2) does not meet requirements of practical desalination applications as it has a very low recovery rate. Such configuration is not efficient as it discards solution

that is already of a lower concentration than the initial feedwater. Here, we further develop TDD with a scalable thermodiffusive separation device known as the Burgers cascade[47], previously shown to enhance thermodiffusive separation in binary gas mixtures[30] but never in liquid systems. The working principle is shown in Fig. 5a. The Burgers cascade is a device with multiple top($C_{low}$)/bottom($C_{high}$) vertical bifurcation and right–left horizontal recombination cells, each essentially acting as a miniature TDU. The possible dimensions and design of the cell is available in the Supplementary Method 7. $M$ and $N$ refer to the total number of rows and columns in the Burgers cascade, respectively, while $m$ and $n$ are the corresponding indices. This complex interconnection of many small channels may be blocked by two-phase formation. However, we demonstrated that bubbles can be avoided by slightly tilting the Burgers cascade and filling it up from the bottom. A miniature Burgers cascade with $M = 5$ and $N = 10$ was manufactured

and the flow structure was visualised with dye as shown in Fig. 5a2. More details of this flow visualisation miniature is available (Supplementary Fig. 12).

We assume that seawater behaves as a binary mixture of ions and water. Based on the MD work, seawater ions are predicted to have an effective $S_T$ 1.8 times larger than that of NaCl/H$_2$O solution. Other assumptions are: mixing of the two streams happens instantaneously at the inlet of each cell; the concentration profile fully develops in each cell; the two streams at the outlet of the cell are perfectly bifurcated at the horizontal mid-plane. After applying our numerical model for a single-pass binary solution to each cell, we can obtain the concentration in each cell of the Burgers cascade. Figure 5b shows the contour plot of concentration in each cell in the Burgers cascade when $\Delta T = 60$ K is applied. The recovery rate can be adjusted by selecting the cut-off location at the Burgers cascade outlet. With 20 cells in each row

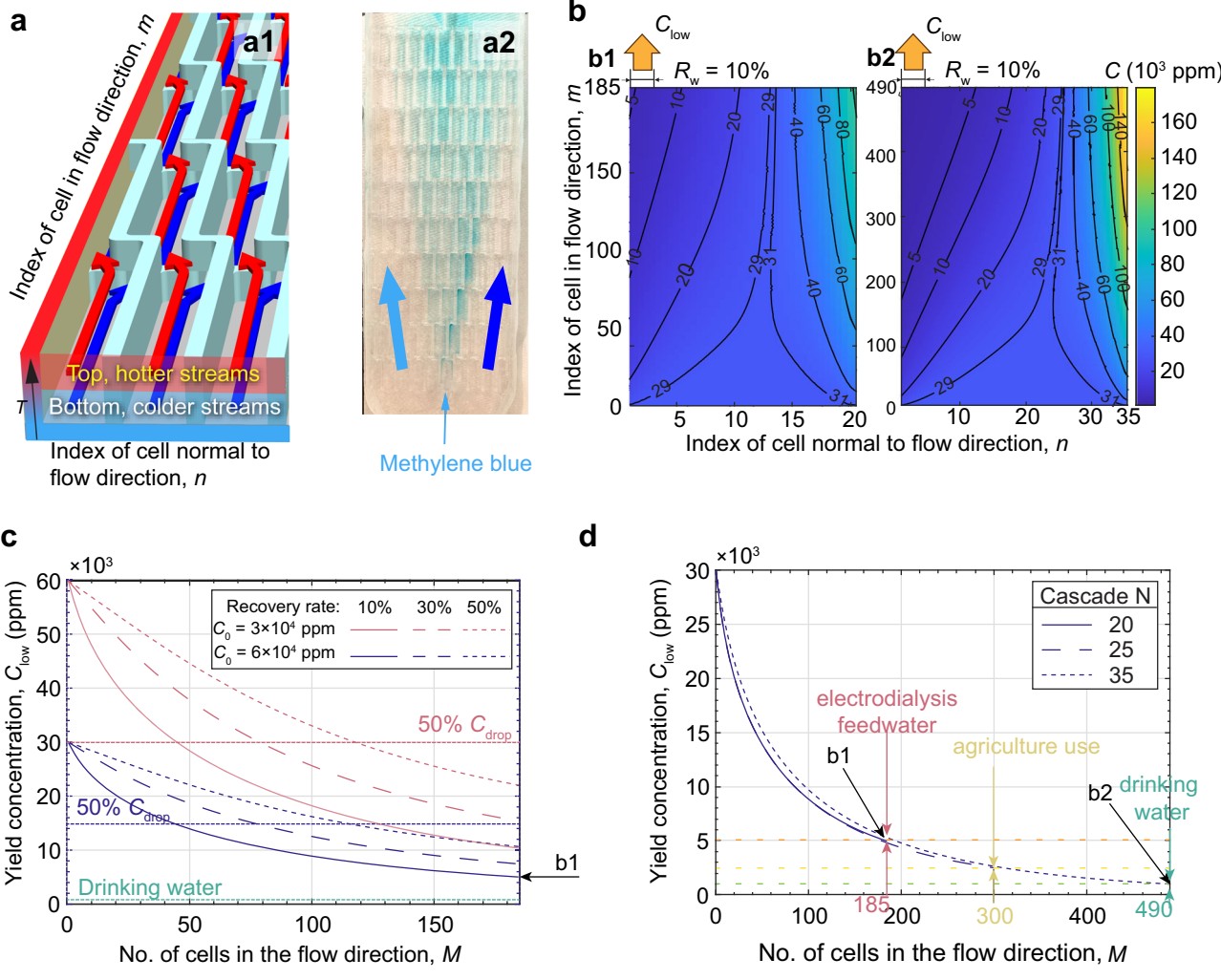

**Fig. 5 | A scalable cascaded TDD device. a** The structure of the Burgers cascade. **a1** The Burgers cascade contains many small cells; each cell is a small parallel-plate channel. In a single cell, the working fluid is divided into two streams, top and bottom, denoted by red and blue arrows respectively. For the entire Burgers cascade, the top is hot and the bottom is cold. The hot stream (red arrow) flows towards the top-left cell outlet and has a lower concentration for thermophobic species, while the cold stream (blue arrow) flows to the bottom-right outlet and has a higher concentration. Water flows in the $m$ direction, where $m$ is a row and $n$ is a column. **a2** Photo of a transparent miniature with $M = 5$ and $N = 10$ showing the lateral spread of dye after injection close to the device inlet. Details of this experiment are provided in Supplementary Fig. 12. **b** Contour plot of the concentration in each cell. $C_0 = 30,000$ ppm, $\Delta T = 60$ K, $S_T$ is assumed to be 1.8 times of

that of NaCl in water. $M = 185$ and $N = 20$ (**b1**) and $M = 490$ and $N = 35$ (**b2**). When the cut-off distance is 10% of the total distance, i.e. the recovery rate $R_w$ is 10%, the yield concentration $C_{low} = 5000$ ppm (**b1**) and 1000 ppm (**b2**), respectively. **c** The concentration of the cold stream is plotted along the flow direction (when $N = 20$) for different recovery rates. Two different feedwater concentrations are considered, $C_0 = 30,000$ and $60,000$ ppm, with the same assumptions as in (**b**) were applied. Drinking water here is defined as water with salt concentration less than 1000 ppm. Lower recovery rate means larger $\Delta C$. **d** The yield concentration when expanding the size of the Burgers cascade fixing $R_w$ to 10%. By expanding the size of the Burgers cascade, seawater can be desalinated to drinking water standard. For different levels of target salinity, the corresponding size $M$ is indicated by coloured labels.

($N = 20$), $C_{low}$ of the yield stream when varying $M$ is plotted in Fig. 5c. With 20 cells in each row and 185 rows, which is comparable to the Burgers cascade with 22 cells in each row and 450 rows in a previously demonstrated experiment[30], the salinity reduction is 50% even when the recovery rate is as high as 50%, for both seawater and brine concentrations.

The Burgers cascade can also be scaled to produce different yield concentrations, as shown in Fig. 5d, with the required number of cells indicated in the legend. Following an analytical method (Supplementary Methods 7 and 8), the pressure drop in the Burgers cascade can be calculated. Even for the case when $M = 490$ and $N = 35$ (Supplementary Fig. 13a), the pressure drop is much smaller than 1 bar, which is what a modest pump can supply and less than 5% of RO pumping requirements[48]. It is important to note that for most designs listed, the pressure drop is less than 10 kPa and the electrical energy consumption to produce 1 $m^3$ of fresh water, when considering only the pump work to overcome the pressure drop, is less than 3 $Wh_e$ (Supplementary Method 8). The minimum thermodynamic limit for the equivalent case[49], i.e. with a feedwater concentration of 30,000 ppm, yield concentration of 5000 ppm and recovery rate of 10%, is 0.4 kWh $m^{-3}$ (Supplementary Method 8). Therefore, the electrical energy consumption of TDD alone is significantly lower than the theoretical minimum energy of separation, only 1% of its value for the aforementioned condition. Moreover, the mass flux equation Eq. (1) dictates that thermodiffusive mass flux monotonically increases with concentration until $C = 0.5$. Combined with the low electrical power consumption of the Burgers cascade, TDD has the potential of being used in a hybrid configuration together with other desalination technologies that are more efficient when the feedwater salinity is low, e.g. RO and ED. When used in conjunction with RO, for example, considering only the electrical energy consumption, the overall energy consumption is reduced by more than 80% from 4.5 kWh$_e$ m$^{-3}$ to 0.7 kWh$_e$ m$^{-3}$ (Supplementary Method 8). TDD may also serve as a pre-treatment method for ED as it is more energy efficient than RO when the feedwater has a salinity lower than 5000 ppm[10,50]. Last but not least, TDD by itself has a comparable salt removal rate as compared to other emerging desalination methods (Supplementary Table 3), justifying its further development beyond the proof of concept presented in this paper.

TDD has interesting trade-offs, such as that between the flow rate $Q$ and the thermal energy consumption (Supplementary Fig. 13b). The concentration profile development time is inversely proportional to the cell height squared, i.e. smaller cell height is preferred to obtain a larger yield (Supplementary Fig. 5c). On the other hand, a larger cell height is preferred to minimise heat flux through the structure because when $\Delta T$ is fixed, the heat flux across the structure is inversely proportional to the channel height (Supplementary Method 7).

## Discussion
### Proof of concept
This work presents a theoretical and experimental framework for implementing thermodiffusion in single-phase thermal desalination applications. We provide modelling results and lab-based measurements that show thermodiffusion can reduce the salinity of seawater. The salinity reduction in a single-pass TDD channel is 450 ppm for a temperature difference of ca. 37 K and mean temperature of 40 °C. Although the salinity drop for a single-pass TDD is small, note the relatively low temperature for operating TDD: this temperature range is ubiquitous as waste heat in industrial settings or hot deserts where water is needed. We experimentally demonstrated that TDD can be scaled up by re-circulating the low-concentration stream through the channel and achieved a salinity reduction (after four passes) of 1200 and 2000 ppm for an initial feedwater concentration of 30,000 ppm (seawater) and 60,000 ppm (brine), respectively. The experimental results are in agreement with our numerical modelling in the continuum regime.

In addition, the thermodiffusive separation in a multi-component ionic aqueous solution was quantified. Due to large errors in ICP spectroscopy, accurate concentration measurements for each ion were not attainable for the multi-component seawater substitute. Nonetheless, our measurements indicate that an effective Soret coefficient is likely to be larger in seawater than in binary NaCl/H$_2$O solutions. Such result agrees with our MD simulation results that show the effective Soret coefficient is larger in multi-ion brine than NaCl brine. The salinity reduction of natural seawater via TDD is therefore expected to be larger than reported values for NaCl/H$_2$O solutions.

We recognise that TDD in the simple parallel-plate channel structure does not seem feasible for practical applications requiring a salinity lower than 5000 ppm and a large recovery rate. As a potential solution to scaling up TDD, we proposed and analysed a multi-channel bifurcation and recombination device called Burgers cascade[30]. Despite the complex internal structure, the pressure drop is not a major concern as the flow rate is very small. We report a theoretical concentration drop greater than 25,000 ppm via thermodiffusive separation using the Burgers cascade with a recovery rate of 10%.

### Advantages
TDD has two major benefits. First, TDD is a thermal desalination process entirely operated in the liquid phase without relying on any functional materials such as membranes or ion-adsorbents that need regular maintenance to avoid extensive materials degradation. It is worth noting that all the TDD experiments reported here were run intermittently for more than one year using the same TDU device (depicted in Supplementary Fig. 1). There was no visible corrosion because the copper blocks were plated in nickel to avoid such surface degradation. From a process perspective, TDD is most similar to ED, where Fickian diffusion mixing is overcome by applying a competing phenomenon, which in this case is thermodiffusion. Second, TDD is driven by relatively low-temperature heat that can be sourced from readily available sources including industrial waste, solar irradiation or simply the surrounding environment. In Fig. 6, a comparison of the specific energy consumption for different desalination technologies is presented. The references corresponding to each numbered item are listed in Supplementary Table 4. We see that thermal-driven methods usually have a much higher energy consumption. Nonetheless, a higher energy consumption can be justified if it is from a source that is readily available in the environment or if it is waste heat (e.g. from industrial or agricultural processes). The use of low-grade thermal energy as the driver for desalination has motivated the development of various desalination technologies including membrane distillation, novel solar-driven distillation and TDD (this work). From an energy perspective, TDD is most similar to membrane distillation where waste heat and large temperature gradients are often implemented. The electrical energy required to produce 1 m$^3$ of 5000 ppm yield water (concentration drop of 25,000 ppm) is less than 3 Wh$_e$ (Supplementary Method 8), which is significantly lower than the minimum thermodynamic limit for desalination under the same recovery and concentration conditions. This indicates that TDD has an outstanding potential in areas where surrounding thermal energy is abundant while access to electricity is limited.

### Challenges and future directions
There are two major challenges that limit the performance of TDD. First, there is a trade-off between yield and heat flux: a smaller height is preferred for a larger yield (Supplementary Fig. 6b) while a larger height is preferred for a smaller heat flux. Second, thermodiffusive separation has a fundamental limit indicated by the mass flux equation Eq. (1). The product $C(1 - C)$ indicates that thermodiffusive mass flux diminishes with decreasing concentration, meaning that TDD is less effective when the feedwater concentration is lower. Considering

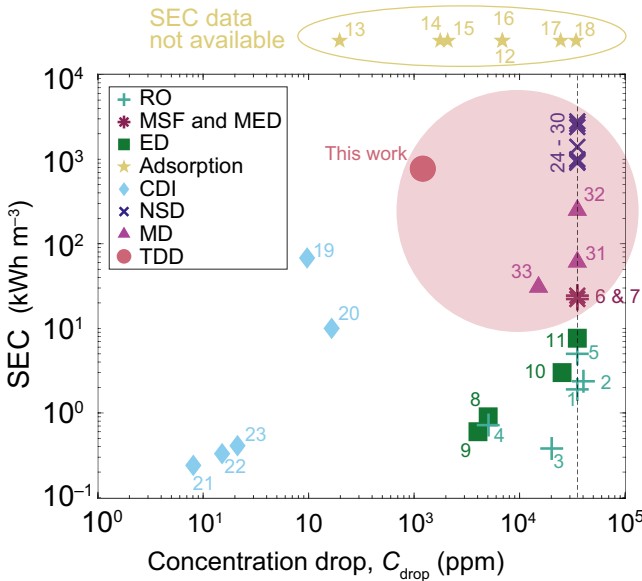

**Fig. 6 | Comparison of different desalination technologies.** The energy consumed per unit volume of yield, or specific energy consumption (SEC), is plotted against the concentration drop for different desalination technologies: reverse osmosis (RO), multi-stage flash (MSF) and multi-effect distillation (MED), electrodialysis (ED), adsorption, capacitive deionisation (CDI), novel solar-driven desalination (NSD), e.g. interfacial evaporation and contactless steam generation, membrane distillation (MD), and the thermodiffusive desalination (TDD) technology proposed in this work. Corresponding sources are indexed and listed in Supplementary Table 4. For some technologies, SEC is dependent on the feedwater salinity and the yield salinity. The vertical black dashed line indicates a concentration drop from seawater to freshwater. The thermal-driven desalination methods are captured by the mauve circle. SEC of different technologies vary vastly and the thermal-driven desalination technologies generally have a much higher energy consumption. For TDD, the presented SEC data is based on multi-pass TDU experiment. A heat flux of 232 W (Supplementary Method 8) was used and the yield was 0.3 L h⁻¹. Under this condition, SEC is calculated as SEC = 0.232 kWh/0.3 L = 773 kWh m⁻³. However, when only considering the electrical energy usage, $SEC_e$, its value drops to nearly 0. SEC data is not available for the adsorption-based desalination.

these challenges, there are several pathways towards the practical implementation of TDD.

First, achieving a better understanding of the fundamental mechanism behind thermodiffusion is paramount to propose strategies that enhance thermodiffusive separation. For example, increasing the isothermal diffusion coefficient shortens the time for the concentration profile to develop in each cell, as the concentration development time is only related to the Fickian diffusion coefficient when the Soret coefficient is small (Supplementary Fig. 5a). Also, increasing the Soret coefficient would enable a greater salinity drop and thus a smaller number of cells is required in the Burgers cascade. Second, as described in Supplementary Method 8, TDD could be used as a pre-treatment method in a hybrid desalination process, concurrently implemented with more mature technologies such as RO and ED, which are more efficient at lower feedwater concentrations. Third, beyond the application of desalination, thermodiffusion can also contribute to the treatment of brine or industrial waste water that contains high-concentrations of pollutants. Thermodiffusive separation is in fact more efficient at salinity levels greater than seawater, as per Eq. (1). A further technology development for TDD would enable an accessible and environmentally friendly desalination method with comparable performance to other desalination concepts recently developed (Fig. 6 and Supplementary Table 3). While still a proof of concept, this paper presents theoretical and experimental underpinnings that may inspire further use of

thermodiffusion as a separation principle in water treatment and other environmental engineering applications.

## Methods

### Simulation of thermodiffusive separation in a channel

Simulations were performed following the same procedure as in our previous work[22]. The channel is a simple parallel-plate channel. The continuum thermodiffusion modelling was based on the conservation of chemical species (NaCl) in a fluid flow with constant velocity field **u** following the governing equation: $\frac{\partial}{\partial t}(\rho C) + \nabla \rho \mathbf{u} C = -\nabla \mathbf{J}$. The essential assumptions are: one-dimensional flow with a known velocity profile (a planar Poiseuille flow), linear temperature profiles across the channel height, and temperature-dependent $S_T$ and $D$[37]. The boundary conditions for temperature are matched to measured temperature values in experiments where twelve thermocouples (six on each side) are placed at the top and bottom boundaries of the 500 mm long channel. The temperature at each grid point is linearly interpolated from the thermocouple readings. The transient change in temperature readings is small and is ignored in the simulation. A fully-implicit finite volume method was used with the discretisation equations available in Supplementary Method 1.

### Sample preparation

NaCl/H₂O samples are prepared on a gravimetric basis using NaCl (Ajax Finechem, Thermo Fisher Scientific, minimum assay 99.7%) and deionised water (Pacific TII 12, Thermo Scientific). The balance has an accuracy of 1 mg (ATX224, Shimadzu). Artificial seawater was prepared using the same apparatus and procedure with rock sea salt sourced from Whyalla, South Australia (Rock sea salt, Olsson's Salt), which was harvested through solar evaporation with no additives.

### Thermodiffusive desalination experiment

In a single-pass experiment, a peristaltic pump (Kamoer FX-STP2) was used to control the volumetirc flow rate $Q$ with an accuracy of 0.1 mL min⁻¹ and a silicone membrane gas exchanger (PDMSXA-1000: PermSelect®1000) connected to −0.8 bar vacuum was used to degas the solution before it enters the channel. The copper blocks that accommodate water bath flow are nickel plated and corrosion was not observed before and after all the experimental runs. To monitor the temperature, six pairs of thermocouples were distributed 100 mm apart on both upper and lower plates of the channel. These K-type thermocouples (410-305, TC Direct) are read out by thermocouple readers (U6-Pro, LabJack). In multiple-pass experiment, between each pass, deionised water was used to flush the channel for around 15 min then compressed air was used to dry the channel for around 15 min to reduce the chances of contamination between solutions of different passes.

### Optical measurement of concentration drop

Phase-shifting interferometry (PSI) is a digital interferometry technique that can resolve the phase difference $\psi$ between test and reference beams. Our PSI technique uses a polarising Mach–Zehnder interferometer. The PSI layout was first developed in[44,51]. The laser (HNL050LB, Thorlabs, wavelength $\lambda$ = 630 nm) is a polarised HeNe laser and a polarising beam splitter divides it into two beams with equal intensity at 90° polarisation angle difference. A quarter-wave plate is placed after the beam recombination to generate circular polarised light. Interferograms are taken at different transmission angles of a linear polariser just before the CCD/CMOS camera (Supplementary Fig. 8a). Here, we used a three-bucket temporal phase-shifting equation. Measuring the phase difference for samples of known $\Delta C$, the relationship between $\psi$ and $\Delta C$ can be established to obtain the contrast factor CF, which relates refractive index variation to concentration variation. CF is dependent on a number of factors including temperature, concentration, species and laser wavelength. Under isothermal condition for a single-wavelength PSI, same $\Delta C$ in

different species has different contribution to $\psi$ and this is why single-wavelength PSI cannot be used to determine the concentration of different species in the solution. Nonetheless, PSI is highly sensitive for detecting concentration differences in binary solutions. More details are available in Supplementary Method 3.

## ICP measurement of ionic concentrations

Concentrations of cations in seawater substitute were measured in two laboratories at *i*) the Australian National University (ANU) using inductively coupled plasma mass spectrometry (ICP-MS, iCap RQ quadrupole, Thermo Fisher) and *ii*) ALS Limited, Environmental Division in Sydney, using inductively coupled plasma atomic emission spectroscopy (ICP-AES). At ANU, samples were gravimetrically diluted using 2% $HNO_3$ with five replicates. The concentrations were drift-corrected using a gravimetrically measured external standard.

## Molecular dynamics simulations

MD simulations were performed by Nanoscale Molecular Dynamics (NAMD) simulation software, following our previous work on non-equilibrium MD framework for assessing thermodiffusion in aqueous electrolytes[52]. A multi-component seawater brine solution was constructed as a model of natural seawater, containing the seven largest monoatomic ions that constitute seawater, that is $Cl^-$, $Na^+$, $Mg^{2+}$, $Ca^{2+}$, $K^+$, $Br^-$ and $Fl^-$. The composition is detailed in Supplementary Table 1. More complex ions were not considered (i.e. $SO_4^{2-}$) due to incompatibility with the simulated water. The water model used is TIP3P-FB. A control NaCl brine solution was constructed with the same molar concentration as the seawater brine, that is approximately 14,000 water molecules and 650 ions. The simulation volume was a rectangular prism of $20 \times 5 \times 5$ nm with periodic boundary conditions, simulating a bulk solution. A quasi-linear temperature profile was maintained across the system by two spatially defined thermostats, spanning a temperature range of 20–60 °C. More details are available in Supplementary Method 6.

Three independent replicates of the NaCl brine and four independent replicates of seawater brine were simulated. After simulation calibration procedures, the ions freely diffused across the temperature axis in a constant volume and energy ensemble for 280 ns. The initial 20 ns is attributed to the time taken for the system to reach quasi-steady state, based on the scaling law[53], and is excluded from final calculations. The time-averaged steady state concentration of ions in each brine solution was measured across the temperature axis with a bin spacing of 1 nm. Since MD is a discrete system, the instantaneous concentration within the small bin volume can fluctuate significantly. Thus, the quasi-steady-state system should be sampled for a sufficient number of times to obtain a statistically meaningful time-averaged concentration. Therefore, the simulations continued to run after 20 ns until 280 ns to allow a long enough time for adequate sampling to be performed. In Supplementary Figs. 9 and 10, it is shown that $260 = 280 - 20$ ns yields a sufficient sample size. Ion concentration drop between the hot and cold boundaries was measured from a linear fit of the concentration profile against average temperature and used to characterise a Soret coefficient for seawater substitute brine and NaCl brine.

## Data availability

The data that supports the findings of the study are included in the main text and Supplementary Information files. Raw data can be obtained from the corresponding author upon request. Source Data file[54] has been deposited in Figshare under accession code DOI: 10.6084/m9.figshare.25054949.

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

## Acknowledgements

This research was funded by the Australian Department of Foreign Affairs and Trade (Grant type: SciTech4Climate), as well as the Foundation for Australia-Japan Studies. We thank Dr Steven Crimp and Dr Mona E. Mahani from the Institute of Climate, Energy and Disasters Solutions at the Australian National University (ANU) for their deep interest in thermodiffusive desalination and for helping us secure funding. We acknowledge the valuable technical advice from Professor Atsuki Komiya and Professor Yuerui Lu. We are grateful to Dr Kasmir Gregory for his interest in thermodiffusive desalination and insightful comments. We thank Mr Reuben Symons for his work in deriving the analytical solution for thermodiffusive separation and Mr Roelof Pottas for designing the thermodiffusive separation unit. The simulations were conducted with resources from the National Computational Infrastructure (NCI). The valuable support from the ANU through an international PhD student scholarship (University Research Scholarship 7382018) is also acknowledged.

## Author contributions

Conceptualisation: S.X., J.F.T. Methodology: S.X., A.J.H., B.C., J.F.T. Investigation: S.X., A.J.H., M.T., B.C., J.F.T. Visualisation: S.X., A.J.H. Funding acquisition: J.F.T. Project administration: J.F.T. Supervision: B.C., J.F.T. Writing—original draft: S.X., J.F.T. Writing—review and editing: S.X., A.J.H., M.T., B.C., J.F.T.

## Competing interests

The authors declare no competing interests.
