## [Peer Review File · Nature Communications]

Thermodiffusive desalinationEditorial Note: Figure 1a in this Peer Review File has been amended to remove third-party material where no permission to publish could be obtained.

REVIEWER COMMENTS

Reviewer #1 (Remarks to the Author):

In this manuscript, the authors studied an interesting method of water desalination. They performed theoretical and experimental studies to support the claims. The information provided in the manuscript is novel. I think Nature Communications is the right publication for this paper. Hence, I recommend accepting the manuscript with a major revision. The authors need to address the following comments and modify the paper accordingly.

1. Line 92 - How did the authors ensure the flow is fully developed?
2. The uncertainty analysis linked to experimentally measured parameters needs to be discussed in detail with equations.
3. The repeatability information of the experimentally measured parameters needs to be added.

Reviewer #2 (Remarks to the Author):

This manuscript describes a method to desalinate salt water by using low-grade heat with a process based on thermodiffusion. It is a proof-of-concept study based on experiments, CFD simulations and MD simulations. The main result is a demonstration of the method's feasibility and efficiency.

Production of fresh water is a growing challenge, and the authors argue convincingly that their concept is an interesting candidate to be further pursued. The authors compare their method and results with existing methods and discuss honestly its advantages and disadvantages. The work is very thorough, well described, and all its aspects are covered. In my opinion, this work is a significant contribution to the literature on the topic of water desalination.

The conclusions are well based on the experimental and simulation results and the statistical and theoretical analyses. The authors check the consistency of analytical and simulation results and demonstrate that the work is of a high standard. The supplementary notes give sufficient details for a reproduction of the results, provided a similar experimental setup is available.

I have found no flaws in this work, but I have some more detailed comments and questions:

1. Line 50: I agree that thermodiffusion is weaker than Fickian diffusion, but this cannot be argued by the value of the Soret coefficient. For one reason, D_T and D have different dimensions. Secondly, Eq. (1) shows that for zero mass flux, the two effects are in balance. I suggest that the authors rephrase or justify their statement.
2. Line 77: Although I do not know of any other work using single-phase thermal desalination, the statement that this is the first is somewhat dangerous.
3. Page 7: I would have liked to see a discussion of Taylor dispersion in this context.
4. Lines 204 and 205: What are super- and subscripts? If they are uncertainties, the magnitude of the

uncertainties do not justify all the digits in the result as significant.

5. Figure S8: The authors claim that steady state is reached after 20 ns, but the figure shows that it is reached at about 175 ns. Please comment.

6. A general question: The separation process will stop if an inversion temperature or composition occurs. In what temperature range, type of salts, and composition ranges may this happen?

Reviewer #3 (Remarks to the Author):

The manuscript presents a thermal diffusion-based method for water desalination. The authors have revisited the well known thermal diffusion or thermo-diffusion or Soret Ludwig effect for which the physics of the phenomenon is well researched. They have proposed a method for desalination of water. The concept had been proposed in past (according to the manuscript; at least theoretically) but implementation has not been attempted or proved to be feasible. The method proposed is simple, yet quite valuable. There are certainly advantages in the proposed method from various angles but more importantly in my view is the fact the utilisation of thermo-osmosis in desalination has been materialised. The paper is well written and comprises the results of both experiments and modelling for the design and validation of the concept. However, there are major points (below) that should be revisited and discussed.

1) Based on my understanding of the work, the solution to utilise the thermo-diffusion desalination which is proposed in this manuscript, works only at a delicate condition where the drag force of the flow (laminar flow) does not disturb considerably the thermo-diffusion effects on ions. In other words, if the flow moves towards a kind of turbulent flow, utilisation of thermo-diffusion is impractical. The manuscript describes that the system was tested and numerically designed for a range of channel thicknesses and flow rates. Figures S5 and S6 are not so helpful to establish a fundamental insight into the hydraulics of the problem. There is a need to discuss the design in the form that provides a clear picture of at what flow regime a thermodiffusion desalination should work or not. For that there is a need to present results based on Reynolds's number or other non-dimensional parameter to describe the balance between forces acting from the flow and also by the thermal gradient. Obviously there are various parameters to be considered. The way it has been presented, would not help to establish how "theoretically" the mystery of utilising the thermo-diffusion in desalination (150 years mystery) has been solved. There should be a new Figure on the main text showing clearly how the problem has been resolved (theoretically). That should be the most interesting figure in my view.

2) Related to the above discussion, I think an additional physical effect is missing in the entire discussions and that is thermo-osmosis. A thermal gradient in a small size channel can create a flow so there should be two forces acting in the same axis (normal to the plates) which are thermo-diffusion and thermo-osmosis. The question is that to what extent such effects are important and why it has not been considered. We know from recent studies that thermo-diffusion and thermo-osmosis can affect each other (i.e. ions can be moved by the fluid flow induced by thermo-osmosis or thermo-osmosis can generate a fluid flow). To what extent such interactions would be important should be discussed in the

context of the concept proposed.

3) There are two minor corrections that may be needed for further clarifications in section 2.1. (i) the term “non-isothermal channel” should be further clarified (e.g. a channel where there is a thermal gradient) and (ii) the sentence “...., which further stabilises the flow[35].” is unclear in terms what the authors mean by “stabilises.

4) The theoretical description of multicomponent ionic diffusion under thermal gradient has progressed and there are cross-diagonal terms in the equations (D12 etc) that reflect the impact of the transport of one ion on the transport of the other one when the diffusion is the dominant flow. Such effects have also been included when concentration and thermal gradients induce mass diffusion. Would such effects (at least theoretically) important in the concept? Page 8 results.

5) Figures 2 (especially 2a) is not useful. I was not able to extract the details from the figures. Suggest 22 to be presented in a larger size.

6) I was a bit confused with the length of the spacer, does that extend through the length of the channel or it is partial at the end (inconsistency between Figs S1 and S2d with Fig 1a).

7) Page 14; advantages: This looks to me the least developed section of the paper; yet the most important from practical side. There should be a Figure/diagram to compare the proposed method (in terms of performance; energy usage or other engineering factors of desalination technologies).

Reviewer 1

R1 In this manuscript, the authors studied an interesting method of water desalination. They performed theoretical and experimental studies to support the claims. The information provided in the manuscript is novel. I think Nature Communications is the right publication for this paper. Hence, I recommend accepting the manuscript with a major revision. The authors need to address the following comments and modify the paper accordingly.

Response:

We thank the reviewer for their time and effort in assessing our manuscript. Their valuable feedback was considered in this major revision. We actioned changes based on three comments from Reviewer 1. We note that other reviewers also requested more information on the flow dynamics in the thermodiffusive desalination channel. Thus, a major revision was conducted based on these feedback.

R1C1 Line 92 - How did the authors ensure the flow is fully developed?

Response:

Thank you for mentioning this important point. In fact, there was a significant amount of thought and analysis put into designing the thermodiffusive desalination unit (TDU). The original submission did not provide enough details on such design process. In response to your comment, first, we modified Fig. 1a to clearly label the entrance length and indicating that the flow is laminar throughout most of the channel. We also added Fig. 1b to better describe the hydrodynamics of the saline solution in the channel. In addition, we now report the calculated hydrodynamic and thermal entrance lengths. Lastly, the revised manuscript now includes a newly added Supplementary Note 2 on the TDU design rationale, along with modified Supplementary Figs. 5 and 6.

Based on the flow speed and channel dimensions, we calculated the Reynolds number and found that the flow is well within the laminar regime for a large range of reasonable channel dimensions. Thus, the hydrodynamic and thermal entrance lengths could be calculated analytically.

Revision:

The hydrodynamic and thermal entrance lengths are now indicated in Fig. 1a. The newly added Fig. 1b better shows the hydrodynamics in the channel. Fig. 1c in the revised manuscript is Fig. 1b of the original submission.

Revised Figs. 1a and 1b with corresponding caption (Fig. 1c not shown)

Line 120 revised manuscript

The flow within the TDU is laminar ($Re < 350$) because of the reduced flow speed required to achieve a nearly fully developed concentration profile at the channel outlet, i.e. a rather long resilience time ($t > 1$ min) within our 0.5 m channel is needed to obtain a nearly complete thermodiffusive separation. In our TDU design with target operation flow rates, the hydrodynamic, thermal, and concentration entrance lengths are around 0.1%, 1%, and 100% of the entire channel length, respectively.

Details on the entrance length calculation can be found in the newly added Supplementary Note 2.

Line 133 revised Supplementary Information

Furthermore, the entrance length can be solved analytically since Re_{dev} is known. The normalised hydrodynamic entrance length is $L_v/L = 0.04Re_{dev}h/L$ [50]. Substituting Eq. (10), it becomes

$$\frac{L_v}{L} = \frac{0.04\rho\pi^2D}{\mu},$$

which is less than 0.1% for the corresponding thermophysical properties of water in the operating temperature range. Furthermore, from the definition of Prandtl number Pr , the normalised thermal entrance length is $L_T/L = PrL_v/L$. Since $Pr < 15$ for liquid water under normal pressure, L_T/L is always smaller than 1%. From the analytical calculation, we see that the normalised entrance lengths for momentum L_v/L and heat L_T/L are both independent of channel dimensions and are less than 1% of the total channel length. Thus, both a fully-developed plane Poiseuille flow through most of the channel and a quasi-linear temperature profile along the channel height are valid assumptions.

R1C2 The uncertainty analysis linked to experimentally measured parameters needs to be discussed in detail with equations.

Response:

Thank you for pointing this out. The uncertainty analysis is very important and hence was conducted when measuring ΔC with our in-house phase-shifting interferometer. The error $\delta\Delta C$ in measuring the concentration difference was calculated through error propagation as the concentration difference is defined as $\Delta C = \frac{\Delta\Psi}{CF \times OP}$.

Revision:

We now explain in the main text the meaning of the error bars and include the accuracy assessment as a new section in Supplementary Note 3.

Line 136 revised manuscript

An accurate measurement of the ΔC is critical in assessing the performance of TDD, and we found that our in-house PSI was much more accurate than commercial salinity meters based on electrical resistance. Supplementary Note 3 and Supplementary Fig. 8 provide details on the PSI measurement and associated uncertainty. The vertical error bars in Fig. 2a,b and Fig. 3a,b depict the errors $\delta\Delta C$ in concentration difference ΔC measurements, and the errors δC_{drop} in concentration drop C_{drop} measurements, respectively. Measurement errors mainly arise when extracting the phase difference from unwrapped phase-shifted data (insets of Fig. 2a) between C_{high} and C_{low} solutions (i.e. those extracted from the collection bottles in Fig. 1c). A typical relative PSI-based error $\frac{\delta\Delta C}{\Delta C}$ is around 10% for the relatively small salinity difference. In contrast, a commercial electrical resistance-based salinity meter may have a relative error exceeding 100% since the measured ΔC is very small.

More details are available in the newly added section “Measurement accuracy” in Supplementary Note 3

Line 226 revised Supplementary Information

To test the accuracy of our device, we prepared six sets of standard solutions of NaCl/H₂O with known concentration differences. The six sets of solutions prepared had a mean concentration of $C_{\text{mean}} = 25\,000$, $30\,000$, and $35\,000$ ppm with $\Delta C/C_{\text{mean}} = 0.02$ and 0.03 . Each standard solution was prepared gravimetrically with a total solution mass of $50\text{ g} \pm 1\text{ mg}$ and a salt mass of around $1.5\text{ g} \pm 1\text{ mg}$. Based on ΔC calculated from the sample preparation process and $\Delta\Psi$ calculated using Eq. (19), CF was calculated as $1.66\text{ g mg}^{-1}\text{ mm}^{-1}$ with an error δCF of $0.05\text{ g mg}^{-1}\text{ mm}^{-1}$.

Thus, we use PSI to measure the concentration difference with the relationship:

$$\Delta C = \frac{\Delta\Psi}{OP CF}.$$

OP is a fixed value as the equipment is unchanged throughout the process. The relative error $\delta\Delta C/\Delta C$ is calculated through error propagation:

$$\frac{\delta\Delta C}{\Delta C} = \left[\left(\frac{\delta\Delta\Psi}{\Delta\Psi} \right)^2 + \left(\frac{\delta CF}{CF} \right)^2 \right]^{0.5}$$

The error in the unwrapped phase difference $\delta\Delta\Psi$ is the standard deviation when $\Delta\Psi$ is obtained for each pair of pixels using Eq. (19) for each individual measurement. The relative error in the contrast factor $\delta CF/CF$ was calculated from six sets of standard samples and is 0.030 . Thus, for a typical $\frac{\delta\Delta\Psi}{\Delta\Psi}$ value that is around 10% , the propagated error for the $\frac{\delta\Delta C}{\Delta C}$ is around 10.5% . This corresponds to the vertical error bars in Figs. 2a,b and 3a,b.

R1C3 The repeatability information of the experimentally measured parameters needs to be added.

Response:

Thank you for your comment. We agree that this information needs to be added.

For the TDU, although the overall concept is simple and straightforward, some care is required to ensure the repeatability. This includes some points already mentioned in our original submission, such as degassing the saline solution, nickel-plating the copper parts to avoid corrosion, and ensuring equal flow rates for the top and bottom streams. Another parameter that can affect the repeatability of the experiment is the channel wall temperature, which may be affected by bubbles accumulating on the heat exchange surface.

Revision:

We added the relevant information in the main manuscript and more detailed explanation in the Supplementary Information.

Line 165 revised manuscript

Bubbles from the water circulation may accumulate at the heat exchange surface (Supplementary Fig. 1b), which reduce the local heat transfer coefficient and hence lower ΔT . Without visible bubbles, the uncertainty in ΔC obtained for the same set water circulation temperatures was within the PSI measurement accuracy, meaning an excellent repeatability was obtained. Despite aiming at having the same wall temperature profiles when investigating the effect of the volumetric flow rate Q , the wall temperature varied due to the different thermal resistance caused by the bubbles. Difference in wall temperature slightly change predictions of ΔC for CFD Case 1 and Case 2, as shown in Fig. 2b when the volumetric flow rate is less than 3 mL min^{-1} .

The bubbles that can affect the heat exchange are shown in a supplementary photo.

Revised Supplementary Fig. 1b
Further explanation is added to the Supplementary Information.

Line 259 revised Supplementary Information

The temperature profile along the channel wall can be affected by bubbles accumulating on the heat exchange surface. Due to buoyancy, bubbles appeared more frequently on the top surface of the cold (bottom) water circulation cavity (as indicated in Supplementary Fig. 1b). For nearly identical wall temperature profiles, the thermodiffusive separation was found to have a good repeatability based on four trials where the set $T_{\text{mean}} = 40\text{ }^{\circ}\text{C}$ and $\Delta T = 50\text{ K}$ yielded $\Delta C = (429 \pm 49)\text{ ppm}$. The error is within the accuracy of our in-house PSI concentration measurement system.

Reviewer 2

R2 This manuscript describes a method to desalinate salt water by using low-grade heat with a process based on thermodiffusion. It is a proof-of-concept study based on experiments, CFD simulations and MD simulations. The main result is a demonstration of the method's feasibility and efficiency.

Production of fresh water is a growing challenge, and the authors argue convincingly that their concept is an interesting candidate to be further pursued. The authors compare their method and results with existing methods and discuss honestly its advantages and disadvantages. The work is very thorough, well described, and all its aspects are covered. In my opinion, this work is a significant contribution to the literature on the topic of water desalination.

The conclusions are well based on the experimental and simulation results and the statistical and theoretical analyses. The authors check the consistency of analytical and simulation results and demonstrate that the work is of a high standard. The supplementary notes give sufficient details for a reproduction of the results, provided a similar experimental setup is available.

I have found no flaws in this work, but I have some more detailed comments and questions:

Response:

Thank you for the positive and constructive feedback. We have considered all of your comments and believe the current version is stronger than the original manuscript.

R2C1 Line 50: I agree that thermodiffusion is weaker than Fickian diffusion, but this cannot be argued by the value of the Soret coefficient. For one reason, D_T and D have different dimensions. Secondly, Eq. (1) shows that for zero mass flux, the two effects are in balance. I suggest that the authors rephrase or justify their statement.

Response:

Thank you for pointing it out. We agree with your comment. We intended to state that the concentration difference induced by thermodiffusion is generally small. However, as the reviewer suggested, a direct comparison of mass diffusion from drivers with different physical quantities (temperature and concentration gradients) is inappropriate.

Revision:

We have now rephrased this explanation.

Line 48 revised manuscript

The Soret coefficient S_T is defined as the ratio between the thermodiffusion and Fickian diffusion coefficients, $S_T \equiv D_T/D$. Based on Eq. (1), ΔC can be approximated as $\Delta C \approx C_0(1 - C_0)S_T\Delta T$ for quasi-linear temperature and concentration profiles with small changes. S_T of NaCl in water is in the order of 10^{-3} K^{-1} [2, 6], meaning the ΔC that can be induced by thermodiffusion is small, limited by the ΔT achievable within the liquid phase of seawater.

R2C2 Line 77: Although I do not know of any other work using single-phase thermal desalination, the statement that this is the first is somewhat dangerous.

Response:

We agree that claiming to be the “first” is delicate because we cannot guarantee that we know all published literature and unpublished work. Hence, we removed this expression or added “to the best of our knowledge” when appropriate, following *Nature Communications* guidelines. In total, there are five changes made.

Revision:

For your reference, the following part in the abstract was modified.

Original abstract

Here, we propose and assess the first thermal desalination method entirely operated in the liquid phase, *i.e.* without evaporation, freezing, membranes or ion-adsorbing materials. The separation principle is based on thermodiffusion, the migration of species under a temperature gradient.

Change 1. We decided to keep the ‘first’ claim in the abstract to reflect our understanding of preceding work and the novelty of our work. However, we added “to the best of our knowledge” to emphasise that this claim is based on an extensive but not exhaustive literature review.

Revised abstract

Here, we propose and assess a thermal desalination method based on thermodiffusion, the migration of species under a temperature gradient. To the best of our knowledge, it is the first thermal desalination method operated entirely in the liquid phase, notably excluding evaporation, freezing, membranes, or ion-adsorbing materials.

Change 2. Regarding the location mentioned in the comment (Line 77), the sentence ‘this study is the first to explore a single-phase thermal desalination process’ was rephrased.

Line 79 revised manuscript

While still a proof of concept, this study provides valid pathways towards realising a single-phase thermal desalination process, demonstrating that water desalination is possible without the need for materials or phase change.

Change 3. We removed ‘the first’ in Line 315 of the original manuscript (Discussion section).

Line 354 revised manuscript

This work presents a theoretical and experimental framework for implementing thermodiffusion in single-phase thermal desalination applications.

Change 4. In Line 326 of the original manuscript, we stated that ‘In addition, for the first time, the thermodiffusive separation in a multi-component ionic aqueous solution was observed.’ However, there was an attempt made by Caldwell and Eide [1] to characterise the thermodiffusion of seawater, but their experiments were not accurate enough to prove that the presence of other ions can affect thermodiffusion of ionic species. Therefore, this sentence was rephrased.

Line 365 revised manuscript

In addition, the thermodiffusive separation in a multi-component ionic aqueous solution was quantified.

Change 5. In Line 340 of the original manuscript, we removed ‘the first’.

Line 378 revised manuscript

TDD has two major benefits. First, TDD is a thermal desalination process entirely operated in the liquid phase without relying on any functional materials...

R2C3 Page 7: I would have liked to see a discussion of Taylor dispersion in this context.

Response:

In the original submission, we considered the possibility of the solute molecules being carried along the streamlines thus the effective diffusivity of the solution along the channel is enhanced while performing the simulation.

Line 32 Supplementary Note 1, original manuscript

For calculation of concentration at the grid interface, we consider the Péclet number (Pe). Pe is defined as the advective transport rate to diffusive transport rate. $Pe \gg 1$ in the x direction, thus an upwind scheme is used. While in the y direction, $Pe = 0$, so a piecewise-linear profile is used. The coefficients at the control volume faces are calculated as the harmonic mean of adjacent grid values.

Nonetheless, we agree this part of the discussion should also be performed for the designed TDU channel. To keep the consistency with the simulation, the discussion is still presented in the form of Pe . The dominant mass transfer mechanisms in different directions are now clearly stated in the main manuscripts. In addition, we have included a discussion on Taylor dispersion in the form of the Péclet number calculation in the newly added Supplementary Note 2.

Revision:

Figure 1b now explicitly labels Pe in the x and y directions and mentions the dominant mass transport mechanism in its caption.

Fig. 1b is also shown in p.2 of this document. In its caption:

Revised Figure 1 caption

b, ...The Péclet number, Pe , indicates an advection-dominant flow in the x direction and diffusion-dominant in the y direction. ...

This information was referred to again in the main text.

Line 125 revised manuscript

We also found that the mass transport is advection-dominant along the channel (Péclet number $Pe_x \approx 10^3$) while mass diffusion dominates across the channel height ($Pe_y = 0$).

The calculations are validated by the good agreement between simulation and experimental results.

Line 188 revised Supplementary Information

In addition, in the x direction, Pe_x is in the magnitude of 10^3 , which means the effect of Taylor dispersion is pronounced and the effective mass diffusion along the channel is enhanced by the shear. The high degree of agreement between experiments and simulations (Figs. 2a,b and 3a,b), which utilise an upwind scheme in the x direction and piecewise scheme in the y direction, also support this assessment.

R2C4 Lines 204 and 205: What are super- and subscripts? If they are uncertainties, the magnitude of the uncertainties do not justify all the digits in the result as significant.

Response:

The superscript and the subscript are errors that arise from the calculation of S_T . We agree that the number of significant digits is not meaningful.

Revision:

We found the comment also applies to our experimental results for multi-ion artificial seawater. Since the experimental results are presented first, we explain the meaning of the superscripts and subscripts here. The following modifications were made in the main manuscript.

Line 233 revised manuscript

S_T was $2_{-1}^{+1} \times 10^{-3} \text{ K}^{-1}$ based on both the ICP-AES and ICP-MS measurement results. The superscripts and the subscripts originates from the process of performing a linear fit to $C_{\text{drop}}(n)$ (Fig. 3c). For the linear fit, the slope is the concentration drop per pass C_{drop} and the intersection at $n = 0$ is the initial concentration C_0 . When assuming a planar Poiseuille flow (Fig. 2d.2), both the concentration difference between boundaries ΔC and the concentration profile $C(y)$ (Fig. 2d.1) can be calculated from C_{drop} (Fig. 2d.3). With the known $C(y)$, S_T can be calculated and the details are in Supplementary Note 5. The superscripts and the subscripts corresponds to the linear fit with the largest and the smallest slope, respectively.

We also corrected the number of significant digits in the revised Supplementary Information. Please refer Supplementary Table 2.

In the main manuscript, we added the explanation of the error bars for molecular dynamics simulations.

Line 270 revised manuscript

To relate the scatter plot as in Fig. 4b with S_T , a linear fit was performed to obtain a continuous concentration profile $C(y)$. S_T is then calculated by finding the value of S_T that renders an analytical solution of $C(y)$ close to the linear fit. The linear fit with the largest and the smallest slope corresponds to the superscript and the subscript, respectively. Details of the analytical process can be found in Supplementary Note 6.

We address the concern about the number of significant digits with the following amendment.

Line 268 revised manuscript

A comparison of the Soret coefficients in Fig. 4c(i) shows that the predicted Soret coefficient of $2.4_{-0.1}^{+0.0} \times 10^{-3} \text{ K}^{-1}$ for seawater brine was $1.8_{-0.2}^{+0.3}$ times larger than the predicted Soret coefficient of $1.4_{-0.2}^{+0.2} \times 10^{-3} \text{ K}^{-1}$ for NaCl brine.

R2C5 Figure S8: The authors claim that steady state is reached after 20 ns, but the figure shows that it is reached at about 175 ns. Please comment.

Response:

In the original Supplementary Fig. S8c (now Supplementary Fig. 9c), the plotted data was the time-averaged ion concentration, i.e. the averaged concentration starting from time $t_0 = 20 \text{ ns}$ and ending at time $t_{\text{final}} = 280 \text{ ns}$, not the ‘instantaneous’ concentration at a particular time t . The cumulative average continues to change even after the system has reached a quasi-steady state, as shown in Supplementary Fig. 9c where there is convergence towards its final value at time $t_{\text{final}} = 280 \text{ ns}$ (as pointed out by the reviewer). The reason for using the cumulative instead of instantaneous concentration as the metric for the steady-state convergence criterion is that a molecular dynamics (MD) simulation is a non-continuous system. Thus, the ‘instantaneous’ concentration is obtained by counting the number of ions within a small region in a given time period. However, this data fluctuates significantly (Fig. 9a) due to the randomness of ion distribution. Thus, obtaining conclusive results is generally not possible when analysing instantaneous data.

It is important to note that there are two factors affecting fluctuations in the time-averaged ion concentration in Supplementary Fig. 9c. The first factor is whether the ions have reached a quasi-steady state condition. The second factor is whether the system has been sampled a sufficient number of times, over a sufficiently large timescale, to produce a meaningful and consistent measurement of the average concentration. In Supplementary Fig. 9c, the running time-averaged ion concentration plateaus (or convergence) after 175 ns, as pointed out by the reviewer. This does not mean 175 ns is the time required for the system to reach a quasi-steady state condition, but implies that after 175 ns the system has been sampled sufficiently to produce a meaningful measurement of the local average concentration. The criterion used to determine that the diffusing ions in the system has reached a quasi-steady state was the theoretical scaling law [4]. The scaling law is based on the theoretical time required for the number of ions to sample the temperature axis via diffusion, predicting 20 ns. In addition, in Supplementary Fig. 9c, we observe that fluctuations in the time-averaged ion concentration have diminished in magnitude by 20 ns (indicated by a vertical dashed line). Therefore, we confirmed the judgement that 20 ns is a reasonable estimate of the time required to reach quasi-steady state. The reported value of the steady-state average ion concentration is

the time-averaged value across the time interval [20, 280] ns, which satisfies both that the system is in quasi-steady state condition (not counting any data before $t_0 = 20$ ns) and that there are a sufficient number of measurements to address the sampling issue (using an interval of data that is larger than 175 ns, $\Delta t = t_{\text{final}} - t_0 = 280 - 20 = 260$ ns, where Δt is the time interval).

Line 475 revised manuscript, Methods

After simulation calibration procedures, the ions freely diffused across the temperature axis in a constant volume and energy ensemble for 280 ns. The initial 20 ns is attributed to the time taken for the system to reach quasi-steady state based on the scaling law [4] and excluded from final calculations. The time-averaged steady state concentration of ions in each brine solution was measured across the temperature axis with a bin spacing of 1 nm. Since MD is a discrete system, the instantaneous concentration within the small bin volume can fluctuate significantly. Thus, the quasi-steady-state system should be sampled for a sufficient number of times to obtain a statistically meaningful time-averaged concentration. Therefore, the simulations continued to run after 20 ns until 280 ns to allow a long enough time for adequate sampling to be performed. In Supplementary Figs. 9 and 10, it is shown that $260 = 280 - 20$ ns yields a sufficient sample size.

Supplementary Figure 9 caption

...The ion concentration profile of NaCl brine converges to a measurable steady state condition. The time-averaged ion concentration between simulation time 20 ns and 280 ns is the concentration reported in the main manuscript. The time taken to reach quasi-steady state t_0 is estimated to be 20 ns by the scaling law [4]. For the each highlighted simulation sub-volume ($x \in [5.5, 6.5]$ nm (dark blue), $x \in [9.5, 10.5]$ nm (light blue), and $x \in [13.5, 14.5]$ nm (mauve)), the cumulative running average ion concentration shows reduced fluctuation by $t_0 = 20$ ns and converges over the simulation time interval $[t_0, t_{\text{final}}]$ to the steady state average ion concentration, meaning that the quasi-steady state system has been sampled for a sufficient number of times....

General comments A general question: The separation process will stop if an inversion temperature or composition occurs. In what temperature range, type of salts, and composition ranges may this happen?

Response:

Thank you for the opportunity to provide some thoughts on this point, as this precise question has been the core research question for another line of work we soon intend to publish. Based on literature and some of our unpublished data, the inversion temperature occurs for most seawater components at a relatively low temperature of less than 25 °C, which is below the operating temperature of the proposed thermodiffusive desalination method. The inversion temperature T_0 is dependent on concentration. T_0 usually increases monotonically from an ultra-dilute condition until a moderate concentration value, typically in the range between 0.3 and 0.5 M (mol L⁻¹), then slowly decreases with concentration until reaching a plateau. T_0 is well documented in the literature for NaCl and KCl aqueous solutions. For NaCl/H₂O, T_0 is around 10 [6] to 12 °C [2] at the concentration of 0.5 M. For KCl/H₂O, T_0 is around 20 °C at 0.5 M [6]. We measured T_0 for other components in seawater and found that for MgCl₂ and MgSO₄, T_0 is lower than 5 and 2 °C, respectively, for a range of concentrations between 0.02 and 2 M. Some preliminary results can be found in the following conference abstract: https://www.researchgate.net/publication/376854108_Optical_measurement_of_thermodiffusion_inversion_temperature_in_binary_solutions_using_digital_interferometry

Revision:

To provide some insights into this matter, we added a brief description in the manuscript.

Line 130 revised manuscript

The mean temperature T_{mean} and temperature difference between walls ΔT were chosen such that the temperature of the bottom wall T_{cold} in the TDU (either single channel or a Burgers cascade) is above the inversion temperature at which thermophobic-thermophilic transition occurs. The inversion temperature has been reported to be between 10 [38] and 12 °C [37] for an aqueous NaCl/H₂O at seawater concentration.

Reviewer 3

R3 The manuscript presents a thermal diffusion-based method for water desalination. The authors have revisited the well know thermal diffusion or thermo-diffusion or Soret Ludwig effect for which the physics of the phenomenon is well researched. They have proposed a method for desalination of water. The concept had been proposed in past (according to the manuscript; at least theoretically) but implementation has not been attempted or proved to be feasible. The method proposed is simple, yet quite valuable. There are certainly advantages in the proposed method from various angles but more importantly in my view is the fact the utilisation of thermo-osmosis in desalination has been materialised. The paper is well written and comprises the results of both experiments and modelling for the design and validation of the concept. However, there are major points (below) that should be revisited and discussed.

Response:

Your positive and constructive feedback are greatly appreciated. The revised manuscript now reflects major changes based on your input.

R3C1(a) Based on my understanding of the work, the solution to utilise the thermo-diffusion desalination which is proposed in this manuscript, works only at a delicate condition where the drag force of the flow (laminar flow) does not disturb considerably the thermo-diffusion effects on ions. In other words, if the flow moves towards a kind of turbulent flow, utilisation of thermo-diffusion is impractical. The manuscript describes that the system was tested and numerically designed for a range of channel thicknesses and flow rates. Figures S5 and S6 are not so helpful to establish a fundamental insight into the hydraulics of the problem. There is a need to discuss the design in the form that provides a clear picture of at what flow regime a thermodiffusion desalination should work or not. For that there is a need to present results based on Reynolds's number or other non-dimensional parameter to describe the balance between forces acting from the flow and also by the thermal gradient. Obviously there are various parameters to be considered. The way it has been presented, would not help to establish how "theoretically" the mystery of utilising the thermo-diffusion in desalination (150 years mystery) has been solved. There should be a new Figure on the main text showing clearly how the problem has been resolved (theoretically). That should be the most interesting figure in my view.

Response:

Thank you for your valuable comment. We acknowledge that this important aspect was not included in the original manuscript. While designing the experiments, we found that the hydrodynamics were rather straightforward. However, this analysis process should not be omitted in our explanation. To amend this point, the revised manuscript now states the transport phenomena characteristics for an effective thermodiffusive desalination process. We now include details on dimensionless numbers, entrance lengths, and the design rationale within a newly added Supplementary Note 2.

Revision:

Three important points were added to the manuscript.

(i) We now mention explicitly that the flow is laminar. Importantly, this is reflected on the new concept figure Fig. 1a,b and its caption.

Line 116 revised manuscript

We recently reported numerical simulation results [7] of the thermodiffusive separation within a parallel-plate channel with a plane Poiseuille flow and a linear temperature gradient. Details of the numerical simulations relevant to this work are available in the Supplementary Information Note 1, along with Supplementary Figs. 2–5. The design rationale for the TDU is detailed in Supplementary Information Note 2, along with Supplementary Fig. 6.

About the developing length, please refer to the reply under R1C1.

In the newly added Supplementary Fig. 5a, the effect of D and S_T on the concentration development time (time constant τ_{th}) is presented. We show that for aqueous electrolytes, the concentration development time τ_{th} can be reasonably well approximated with an empirical solution.

Supplementary Figure 5

Supplementary Figure 5 | Effect of different parameters on thermodiffusive separation. **a**, The effect of the D and S_T on the thermodiffusion time constant τ_{th} , i.e. the time required for the concentration profile to reach steady state. For most species where $S_T < 0.1 K^{-1}$ (includes seawater), τ_{th} is only dependent on the isothermal diffusion coefficient D . Moreover, our simulation results show that $\tau_{th} \propto D^{-1}$, agreeing with Eq. (8). In contrast, for $S_T > 0.1 K^{-1}$, the thermodiffusion process starts to exhibit an effect on the time response as per observed τ_{th} dependence on S_T .

In the newly added Supplementary Fig. 6a, the Reynolds number when a fully developed concentration profile is achieved as a function of thermodiffusive desalination channel dimensions (height and length) is plotted. It was found that *the flow is always laminar* based on the flow speed limit requirements for the concentration development time.

Supplementary Figure 6

Supplementary Figure 6 | TDU design considerations. **a**, Reynolds number, Re , as a function of the TDU channel length and height when the flow speed of the NaCl/ H_2O fluid in the channel is limited by the thermodiffusion time constant, τ_{th} .

(ii) We better describe mass transport mechanism in the TDU in the new concept figure Fig. 1b and in the revised manuscript. Please refer to the reply under R2C3.

(iii) The effect of the positive temperature gradient (hotter on top) on stabilising the flow via thermal and species (concentration) stratification is briefly discussed.

Line 98 revised manuscript

In thermophobic transport, ionic species tend to accumulate towards the cold bottom side, resulting in a negative concentration gradient. Along with the negative density gradient brought by thermal stratification, species stratification ensures that natural convection-induced mixing does not occur in our thermodiffusive separation channel [35].

Details on the above key points are presented in the revised Supplementary Information in the newly added added Supplementary Note 2. This technical note explaining the hydrodynamics and the design process for a thermodiffusive desalination channel.

R3C1(b) Related to the above discussion, I think an additional physical effect is missing in the entire discussions and that is thermo-osmosis. A thermal gradient in a small size channel can create a flow so there should be two forces acting in the same axis (normal to the plates) which are thermo-diffusion and thermo-osmosis. The question is that to what extent such effect are important and why it has not been considered. We know from recent studies that thermo-diffusion and thermo-osmosis can affect each other (i.e. ions can be moved by the fluid flow induced by thermos-osmosis or thermo-osmosis can generate a fluid flow). To what extent such interactions would be important should be discussed in the context of the concept proposed.

Response:

We thank the reviewer for this detailed comment. After careful inspection, it seems that there was a misunderstanding of the proposed separation scheme stemmed from the original Fig. 1a (shown below as a reference). Previously we had a ‘separation’ label along a thin horizontal plane, which could have been misinterpreted as a physical membrane. We apologise for this possible misunderstanding.

The “separation” in the original Fig. 1a actually referred to thermodiffusive separation in a free convection environment, i.e. without membranes. The body of saline water that is flowing through the TDU is a continuous fluid flow only separated into two streams at the end of the channel by the spacer, as in the TDU photo Supplementary Fig. 1b. The spacer is an impermeable metallic material that cannot filtrate species. Therefore, it is not responsible for establishing any concentration gradient and is only there to bifurcate the laminar fluid flow in the channel into two streams for further processing. Once the separation is fully developed, there is no mass transfer normal to the plates as these form a hard boundary to the fluid in the channel.

To the best of our knowledge, thermo-osmosis can only occur between two body of fluids held at different temperatures *and* separated by a semi-permeable membrane (or other porous separation). Thermo-osmosis has been reported in nano-channels [3] and porous media [5]). However, if the reviewer is aware of any papers where thermo-osmosis is reported for a continuous fluid without membranes, then we are more than happy to further revise the manuscript.

Revision:

We modified the concept figure Fig. 1a to avoid confusion and further clarify our water desalination scheme (without membranes).

R3C2 There are two minor corrections that may be needed for further clarifications in section 2.1. (i) the term “non-isothermal channel” should be further clarified (e.g. a channel where there is a thermal gradient) and (ii) the sentence “. . . , which further stabilises the flow [35].” is unclear in terms what the authors mean by “stabilises.”

Response:

We modified the sentences as suggested for clarification.

Revision:

Line 88 revised manuscript

A multi-component salt solution passes through a rectangular channel with a vertical positive temperature gradient (heated from the top), as shown in Fig. 1b.

The second point is modified as in R3C1(a) Point 3.

Line 98 revised manuscript

In thermophobic transport, ionic species tend to accumulate towards the cold bottom side, resulting in a negative concentration gradient. Together with the negative density gradient brought by thermal stratification, species stratification ensures that natural convection-induced mixing does not occur in our thermodiffusive separation channel [35].

R3C3 The theoretical description of multi-component ionic diffusion under thermal gradient has progressed and there are cross-diagonal terms in the equations (D12 etc) that reflect the impact of the transport of one ion on the transport of the other one when the diffusion is the dominant flow. Such effects have also been included when concentration and thermal gradients induce mass diffusion. Would such effects (at least theoretically) [be] important in the concept? Page 8 results.

Response:

Thank you for this comment. As pointed out by the reviewer, cross-diffusion effects are important considerations when it comes to the analysis of diffusion in ternary or higher order multi-component solutions. We had included the discussion on how we treated the cross-diffusion coefficients in Lines 170–183 of Supplementary Note 4 in the original submission. However, the reasoning behind the simplification approach was not clear. To summarise, first, a strong electrostatic coupling that results in different ions in seawater moving collectively as a single species is likely to occur based on molecular dynamics simulation and the ICP results (though with large errors). Thus, we approximate the multi-component solution that mimics seawater as a binary ionic aqueous solution, i.e. with an effective ‘ion’ and water as its constituents. This is a valid approximation as we are only interested in the overall removal of ions from the saline solution, not the behaviour of any individual ions. Second, characterisation of the cross-diffusion coefficients requires accurate measurement of the concentration of each ion species. However, our single-wavelength interferometer (Supplementary Fig. 8a) cannot distinguish between different ions and ICP methods are not accurate enough. Thus, it is not possible to resolve all the cross-diffusion coefficients.

Revision:

The reasoning behind not directly calculating the cross-diagonal terms in the diffusion equation is now more clearly explained in Supplementary Note 5.

Line 300 revised Supplementary Information

We prefer the simplification of treating seawater as a binary mixture of water and ions because we are only interested in the overall removal rate of all the ions instead of individual ion behaviour. Through molecular dynamics simulation, we found that all species in the seawater move collectively, as has been pointed out in the section ‘TDD at the molecular level’. We believe the likely cause is the strong electrostatic coupling in ionic electrolyte, which makes it different from the thermodiffusive behaviour of some hydrocarbon mixtures where one component show thermophilicity while the other two show thermophobicity [53]. Therefore, seawater can be effectively treated as binary mixture of ions and water.

Another reason for simplification is that the calculation of these cross-diffusion coefficients rely heavily on the accurate quantification of the concentration of each species in the solution. However, it is not quite possible to perform this task for seawater. For aqueous electrolytes, its composition cannot be viewed as NaCl, MgSO₄ etc. but as Na⁺, Cl⁻, Mg²⁺, SO₄²⁻ because the ionic bounds are broken and the cations and anions are separately surrounded by water molecules. This means seawater is essentially a twelve-component (or even more) mixture. ICP-OES or ICP-MS can only roughly the concentration of cations and we won’t have knowledge of the anion concentrations. In addition, the working principle for the ICP methods is that it measures the intensity of the flame at different wavelengths so that the concentration of different cations can be determined. For seawater, Na⁺ is overwhelming abundant, making measuring the slight changes in the light intensity at other wavelengths difficult. Despite many communications and iterations with the ICP lab on campus, we could not achieve a better accuracy with the ICP measurements. That means the method of physical extraction then measure the sample concentrations is not possible. The other way is to use non-intrusive optical method. However, two-wavelength system is need for ternary solution [54] and for each component added, a different wavelength has to be added to the optical system. Moreover, there is the exponentially increasing workload that’s related to contrast factor measurement [54]. So far, to the best of our knowledge, when it comes to thermodiffusion, the cross-diffusion coefficients are only calculated for ternary solution in published data and there is a group trying to characterise quaternary solutions with a three-wavelength laser. For a twelve-solution mixture, we cannot quite think of possible ways of arranging eleven different lasers on an optical table and we believe at this point of time, it is not possible to extract the cross-diffusion coefficients for each component in seawater.

R3C4 Figures 2 (especially 2a) is not useful. I was not able to extract the details from the figures. Suggest 22 to be presented in a larger size.

Response:

We agree that the original figure failed to scale the size of each panel proportionally with the density of the image. However, we consider Figure 2a to be a very important figure because it trends indicate how separation is enhanced by the temperature difference and higher mean temperatures. The figure was amended to improve readability. Please also note that all source data for plots will be made available to the readers together with the paper.

Revision:

We re-plotted Fig. 2 and hope the new figure is now easier to read.

R3C5 I was a bit confused with the length of the spacer, does that extend through the length of the channel or it is partial at the end (inconsistency between Figs S1 and S2d with Fig 1a).

Response:

We are sorry for the confusion caused. ‘Separation’ in the original Fig. 1a does not refer to any physical separation such as a semipermeable membrane. It referred to the thermodiffusive ‘separation’ process, i.e. the heterogeneous concentration profile caused by thermodiffusion. The impermeable spacer is a pointy shim that is only present at the outlet of the channel (occupying a small length) and with the purpose of bifurcating the flow.

Revision:

Fig. 1a was re-plotted to avoid this misunderstandings.

R3C6 Page 14; advantages: This looks to me the least developed section of the paper; yet the most important from practical side. There should be a Figure/diagram to compare the proposed method (in terms of performance; energy usage or other engineering factors of desalination technologies).

Response:

Thank you for this very important point. We acknowledge that the comparison of the proposed thermodiffusive desalination method with other methods is an essential part of the paper. In the original manuscript, we included the comparison as Supplementary Table 3. However, to better highlight the comparison with other methods, a more thorough information gathering and assessment was conducted. We now summarise the performance of different desalination technologies in a new figure in the main manuscript.

Revision:

Please refer to the newly added figure, Fig. 6 for the comparison of thermodiffusive desalination and other methods.

Newly added Fig. 6
Fig.6 | Comparison of different desalination technologies. The energy consumed per unit volume of yield, or specific energy consumption (SEC), is plotted against the concentration drop for different desalination technologies: reverse osmosis (RO), multi-stage flash (MSF) and multi-effect distillation (MED), electrodialysis (ED), adsorption, capacitive deionisation (CDI), novel solar-driven desalination (NSD), e.g. interfacial evaporation and contactless steam generation, membrane distillation (MD), and the thermodiffusive desalination (TDD) technology proposed in this work. Corresponding sources are indexed and listed in Supplementary Table 4. For some technologies, SEC is dependent on the feedwater salinity and the yield salinity. The vertical black dashed line indicates a concentration drop from normal seawater to freshwater. The thermal-driven desalination methods are captured by the mauve circle. SEC of different technologies vary vastly and the thermal-driven desalination technologies generally have a much higher energy consumption. For TDD, the presented SEC data is based on multi-pass thermodiffusive desalination in the TDU. A heat flux of 232 W (Supplementary Note 8) was used and the yield was 0.3 L h^{-1} . Under this condition, SEC is calculated as $SEC = 0.232 \text{ kWh}/0.3 \text{ L} = 773 \text{ kWh m}^{-3}$. However, when only considering the electrical energy usage, SEC_e , its value drops to nearly 0. SEC data is not available for the adsorption-based desalination.

The discussion of the new figure was also added.

Line 384 revised manuscript

In Fig. 6, a comparison of the specific energy consumption for different desalination technologies is presented. The references corresponding to each numbered item are listed in Supplementary Table 4. We see that thermal-driven methods usually have a much higher energy consumption. Nonetheless, a higher energy consumption can be justified if it is from a source that is readily available in the environment or if it is waste heat (e.g. from industrial or agricultural processes). The use of low-grade thermal energy as the driver for desalination has motivated the development of various desalination technologies including membrane distillation, novel solar-driven distillation and TDD (this work).

References in this Response file

- [1] D. R. Caldwell and S. A. Eide. “Separation of seawater by Soret diffusion”. In: *Deep Sea Research Part A, Oceanographic Research Papers* 32.8 (1985), pp. 965–982. ISSN: 01980149.
- [2] Douglas R. Caldwell. “Thermal and Fickian diffusion of sodium chloride in a solution of oceanic concentration”. In: *Deep-Sea Research and Oceanographic Abstracts* 20.11 (1973), pp. 1029–1039.
- [3] Wei Qiang Chen, Andrey P. Jivkov, and Majid Sedighi. “Thermo-Osmosis in Charged Nanochannels: Effects of Surface Charge and Ionic Strength”. In: *ACS Applied Materials and Interfaces* 15.28 (2023), pp. 34159–34171. ISSN: 19448252.
- [4] Alejandro Diaz-Marquez and Guillaume Stirnemann. “In silico all-atom approach to thermodiffusion in dilute aqueous solutions”. In: *The Journal of Chemical Physics* 155.17 (2021), p. 174503.
- [5] Roberto Piazza. “Thermal forces’: Colloids in temperature gradients”. In: *Journal of Physics Condensed Matter* 16.38 (2004). ISSN: 09538984.
- [6] Frank Römer et al. “Alkali halide solutions under thermal gradients: Soret coefficients and heat transfer mechanisms”. In: *Journal of Physical Chemistry B* 117.27 (2013), pp. 8209–8222.
- [7] Shuqi Xu et al. “Scaling up thermodiffusive separation through a microchannel”. In: *Proceedings of the 23rd Australasian Fluid Mechanics Conference*. 2022, p. 438.

REVIEWERS' COMMENTS

Reviewer #1 (Remarks to the Author):

I would recommend that the paper is now fit to be published.

Reviewer #2 (Remarks to the Author):

The revised MS answers all my questions and comments in a satisfactory way.

Reviewer #3 (Remarks to the Author):

The authors have addressed my comments adequately and the response is comprehensive and clear. Would recommend the paper for publication.